# Brittleness of Concrete under Different Curing Conditions

**DOI:** 10.3390/ma14247865

**Published:** 2021-12-19

**Authors:** Shuai Zhang, Bing Han, Huibing Xie, Mingzhe An, Shengxu Lyu

**Affiliations:** 1School of Civil Engineering, Beijing Jiaotong University, Beijing 100044, China; shuaizhang@bjtu.edu.cn (S.Z.); hbxie@bjtu.edu.cn (H.X.); mzhan@bjtu.edu.cn (M.A.); 16121078@bjtu.edu.cn (S.L.); 2Key Laboratory of Safety and Risk Management on Transport Infrastructures of Ministry of Transport, Beijing 100044, China

**Keywords:** concrete, curing condition, loading age, fracture property, brittleness

## Abstract

In order to shorten construction periods, concrete is often cured using steam and is loaded at an early age. This changes the performance and even the durability of the concrete compared to concrete that has been cured under normal conditions. Thus, the pattern and the mechanism of concrete performance change under different curing conditions, and loading ages are of great significance. The development of brittleness under different curing conditions and loading ages was studied. The evaluation methods that were used to determine concrete brittleness were expounded. Steam, standard, and natural curing conditions were carried out on single-side notched concrete beams as well as on a concrete prism and cubic blocks. The compressive strength and splitting tensile strength of the concrete blocks along with the fracture performance of the concrete beams were tested after 3, 7, 28, and 90 days. The steam curing condition significantly improved the strength of concrete before 28 days had passed, and the standard curing condition improved the strength of concrete after 28 days. Based on the experimental fracture parameters, a two-parameter fracture model was applied to study the development of fracture toughness $K_{\mathrm{IC}}^{S}$, critical crack tip opening displacement ${\mathrm{CTOD}}_{c}$, and critical strain energy release rate $G_{\mathrm{IC}}^{S}$ with hydration age under different curing conditions. With respect to long-term performance, the standard curing condition was better at resisting concrete crack propagations than the steam curing condition was. The characteristic length $l_{\mathrm{ch}}$ and the material length Q under the three curing conditions and the long-term development of brittleness in the concrete indicated that steam curing increased the concrete brittleness. Considering the effects of the curing condition and the loading age, a time-dependent concrete fracture toughness model was established, and the predicted value of the model was verified against the measured value. The results indicated that the model was able to accurately predict the fracture toughness with an error rate of less than 16%.

## 1. Introduction

In order to accelerate the construction progress, steam curing is often applied in order to shorten the curing time so that concrete can be loaded at early ages. However, the steam curing condition exacerbates the hydration reaction of concrete in a short period of time during the early curing stages [1,2,3,4]. Steam curing promotes the development of concrete strength at an earlier age, advances the concrete loading time, and reduces the construction cost. However, steam curing may affect the long-term performance of concrete. Loading at an early age causes micro damage to the internal structure of concrete [5,6,7,8]. The above measures affect the long-term performance of concrete, resulting in increased brittleness and decreased durability [9,10].

Concrete is recognized as a quasi-brittle material. When brittle concrete has been used in engineering infrastructure, sudden structural failures can occur. The brittleness of concrete plays a critical role in many performance indexes. Brittleness not only reflects the internal characteristics of the material but also affects the actual performance of concrete structures [11,12]. Consequently, analyzing the fracture mechanics of concrete is important for studying the brittleness of concrete. Kazemian F., Rooholamini H. and Hassani A. [13] have suggested that the future design of concrete structures will be based on the fracture mechanics of concrete. However, because the application of concrete fracture mechanics has only been implemented in the past 50 years and due to the complexity of concrete materials, researchers have not yet formed a clear understanding of its fracture performance. A study on the brittleness of concrete under different curing conditions and loading ages is therefore of great significance to the design of concrete structures and for quality control in concrete engineering.

Under steam curing conditions, the initial strength of concrete increases; this is because as the curing temperature increases, the degree of hydration during the initial stage is intensified, resulting in the rapid accumulation of numerous hydration products, forming high initial strength [14,15,16]. However, due to initial high-temperature curing, the large number of hydration products that is generated during the early stages of hydration and the hydration reaction time being too short for orderly precipitation results in the formation of a disordered and relatively loose structure, which ultimately results in a loss in the ultimate strength of the concrete [17,18]. Taerwe L. and Schutter, G.D. [19] believed that a thick hydration product film rapidly forms around cement particles, hindering contact between the water and the cement particles, thus increasing the porosity and reducing the hydration rate. When applying steam curing, the hydrated calcium silicate (C-S-H) gel become denser, and the pores are not filled effectively, resulting in increased porosity and an uneven concrete structure after hardening, leading to lower overall strength.

In this study, the strength (compressive strength, splitting tensile strength, initiation load, and load limit), fracture property (fracture toughness $K_{\mathrm{IC}}^{S}$, critical crack tip opening displacement ${\mathrm{CTOD}}_{c}$, and critical strain energy release rate $G_{\mathrm{IC}}^{S}$), and brittleness (material length Q and characteristic length $l_{\mathrm{ch}}$) of concrete under steam curing, standard curing, and natural curing conditions were examined at different loading stages. Then, based on a comprehensive analysis of the effects of the curing conditions and the loading ages on the brittleness of the concrete, a time-dependent concrete fracture toughness model was established.

## 2. Experiment

### 2.1. Experimental Aim

Specimens consisting of single-sided notched concrete beams and cubic and prism concrete blocks were prepared. The specimens were divided into three groups: steam curing, standard curing, and natural curing. The duration for steam curing was three days, and for standard and natural curing, it was 28 days. After curing, each specimen was loaded with 40% of the three-day-old strength. Compressive strength, splitting tensile strength, and fracture tests were all conducted after 3, 7, 28, and 90 days.

### 2.2. Specimen Preparation

Ordinary Portland cement (OPC) was used in this research; the cement density was 1450 kg/m^3^, and its physical properties and chemical components are shown in Table 1 and Table 2, respectively. Crushed stone, consisting of 30% 5–10 mm and 70% 16–25 mm sized particles, was used as the coarse aggregate (CA), and the density of the coarse aggregate was 2750 kg/m^3^. A fine aggregate (FA), which consisted of medium sand with a size of 0.08–2 mm and a fineness of 2.6, was used; the density of the fine aggregate was 2584 kg/m^3^. The chemical components of the fly ash are shown in Table 3. A polycarboxylate superplasticizer (SP) with a water reduction rate of 25% was used in this research. The slump value of the mixtures was 170 mm. The 50 MPa concrete mix proportions from the Zoucheng beam fabrication yard, the same concrete mix that was used to cast the box girders in the Beijing–Shanghai High-Speed Railway project, were used, as shown in Table 4.

Three curing conditions were applied, where the group A was steam-cured, group B was standard-cured, and group C was naturally cured. The second number after the group name denoted the loading age. In the A group, the specimens were further divided into a 3-day-loading group, which was denoted by A-3, a 7-day-loading group, which was denoted by A-7, and a 28-day-loading group, which was denoted by A-28. The specimens in groups B and C were loaded on the 28th day, and the specimen groups are shown in Table 5. The concrete specimens were created according to the GB/T 50082-2002 standard [20].

For the steam curing, after the concrete specimens were created and had been set aside for four hours, the temperature was increased at a rate of 10 °C/h for three hours until the steam temperature reached a constant temperature of 45 °C. The concrete temperature at the core of the beam was controlled so that it did not exceed 60 °C. A constant temperature of 45 °C was maintained for 62 h, and then a controlled cooling rate was implemented at 10 °C/h for three hours. When the specimens were removed from the steam curing box, the temperature difference between the concrete core and the surface was controlled. The temperature between the surface and the environment was also controlled to be within 15 °C. For the standard curing process, the concrete specimens were placed in the standard curing room, the standard curing temperature was set to 21 °C, the humidity was set to 95%, and the curing duration was 28 days. For the natural curing condition, the specimens were covered with geotextile and were sprayed with water, and the temperature was maintained at 14.5 °C, with a curing time of 28 days.

### 2.3. Methods

#### 2.3.1. Compressive and Splitting Tensile Strength Tests

Cubic specimens with the dimensions of 100 × 100 × 100 mm^3^ were used to test the strength of the concrete. Three cubic specimens were used for each group, and the total number for each test was 51. The compressive strength and the splitting tensile strength tests were performed using a universal testing machine in accordance with the GB/T 50081-2002 standard [20], as shown in Figure 1 and Figure 2.

#### 2.3.2. Elastic Modulus Test

Prism specimens with the dimensions of 100 × 100 × 300 mm^3^ were used for the elastic modulus test, which was performed on a universal testing machine in accordance with the GB/T 50081-2002 standard [20], as shown in Figure 3 and Figure 4.

#### 2.3.3. Fracture Property Test

Precast notched-beam specimens with the dimensions of 100 × 100 × 515 mm^3^ were used for the fracture property tests. A total number of 51 beams were used, with 3beams per group. The initial seam height ratio was 0.3, as shown in Figure 5, and the beam was cast with a plastic mold. Before casting, a T-shaped steel plate with a height of 30 mm and thickness of 3 mm was fixed in the middle of the beam span to form precast cracks, as shown in Figure 6.

The three-point bending capacity $F_{3}$ of the precast notched beam after three days of steam curing was measured, and the bending moment $M_{3}\;$in the middle of the beam was calculated. The middle section of the beam was loaded $F_{M}$ after 3, 7, and 28 days, and the bending moment, which was caused by $F_{M}$ in the middle section of the beam, which was 40% of $M_{3}$, was controlled.

The fracture property test was performed by a microcomputer-controlled electronic universal testing machine, crack opening displacement was tested by an electronic extensometer, and fracture strain was tested using a strain acquisition instrument. The applied pressure and the displacement of the beam were automatically recorded by the acquisition module of the testing machine. The fracture property test was repeated at the 3rd, 7th, 28th, and 90th days.

The testing process consisted of loading–unloading–loading. First, a load was applied that was 95% of the maximum load, Pmax, at a rate of 0.02 mm/min. Then, it was unloaded at a rate of 30 N/s. Finally, it was reloaded to failure, as shown in Figure 7 and Figure 8.

The concrete crack initiation was monitored using an electrical method. When the concrete beam was loaded, the strain on both sides of the crack tip increased as the load increased. When the cracks appeared, the concrete that was under tension on both sides of the cracks shrank because of the cracks, and strain hysteresis appeared, and the strain dropped rapidly. However, at this point, the load did not reach the peak value, and the applied load continued to increase, and the strain continued to decrease. The strain hysteresis point was the crack initiation load, and the peak load was the load limit. The strain gauge distance was 30 mm, and the spacing was 10 mm. The test setup and arrangement are shown in Figure 9 and Figure 10.

## 3. Results and Analysis

### 3.1. P-CMOD Curves

The test results were labeled according to the following rules: The first letter represented the curing condition, with A representing steam curing, B representing standard curing, and C representing natural curing; the second number represented the loading age; and the third number denoted the age of the test. For example, A-28-90 represented the test results of the steam-cured concrete loaded at the 28th day and tested at the 90th days. Each group consisted of three beams, and P-CMOD curves were drawn, as shown in Figure 11.

The P-CMOD curves of the concrete beams under no load, that were tested on the 28th day, that were tested under three curing conditions, are shown in Figure 12. On the 28th day, the bending strength of the cured concrete, from the largest to the smallest, was as follows: standard curing > steam curing > natural curing. The results showed that steam curing improved the initial strength of the concrete but resulted in a weakening effect during the later stages of concrete strength growth. This was due to the large number of hydration products that were generated during the early curing stages, when the curing temperature was higher. However, the generated hydration products overlapped and precipitated in a disorderly manner, resulting in increased porosity. The dense hydration products wrapped around the un-hydrated cement particles, which further hindered hydration, resulting in low degrees of final hydration. In addition, the initial slopes of the P-CMOD curves for the steam and standard curing conditions were higher than they for natural curing, indicating that the bending stiffness of the beams during steam and standard curing was higher than it was during natural curing. This occurred because the elastic modulus of the naturally cured concrete was lower, which was consistent with the results from [21,22,23].

The P-CMOD curves of the concrete that had been cured using the standard method were fuller compared to the steam- and naturally cured specimens, and the bearing capacity did not decrease rapidly after reaching the load limit. This phenomenon indicated that the fracture energy of the concrete under standard curing was larger and could consume more energy, resulting in reduced brittleness.

### 3.2. Crack Initiation Load and Load Limit

The crack initiation load and the load limit of the concrete under the three curing conditions are shown in Table 6. After three days, both the crack initiation load and the load limit of the steam-cured specimens were the largest, and the corresponding values for the naturally cured concrete were the smallest. On the 7th day, the data were similar to that of the specimens that had been cured for three days. On the 28th day, the crack initiation load and the load limit of the standard-cured specimens were the largest, slightly larger than that of the steam-cured specimens, and the corresponding values under the naturally cured condition remained the smallest. At the 90th day, under the same conditions (loading at the 28th day), the load limits of the standard-cured specimens were the highest, followed by the steam-cured specimens and the naturally cured specimens.

The crack initiation and load limit data for the specimens under the three curing conditions after 3, 7, 28, and 90 days showed that the steam curing condition significantly improved the early bearing capacity, which adversely affected the strength values in the later stages. Furthermore, the standard curing condition positively affected the later bearing capacity, and the effects of natural curing on the bearing capacity of the concrete during the initial and later stages were weaker than the abovementioned two curing conditions.

The ratio of the crack initiation load to the load limit represented the brittleness of the concrete. At the 28th day, under the same conditions (no loading), the $P_{\mathrm{ini}}$/$P_{\max}$ ratio of the naturally cured concrete was the largest, followed by the steam-cured concrete, and the corresponding value of the standard-cured concrete was the smallest. On the 90th day, the data trend was similar to that from the 28th day under the same conditions (loading at the 28th day).

Based on the $P_{\mathrm{ini}}$/$P_{\max}$ ratio data of the specimens under the three curing conditions, for days 28 and 90, and compared to the standard curing condition, the steam curing condition significantly improved the brittleness of the concrete.

### 3.3. Compressive Strength and Splitting Tensile Strength

The compressive and splitting tensile strength test results of the blocks that had been loaded on the 28th day under the three curing conditions are shown in Table 7 and Figure 13 and Figure 14. Compared to the compressive strength of the blocks on the 3rd day, the compressive strength of the steam-cured concrete increased by 7.11%, 9.65%, and 12.99%, and the corresponding values of the standard-cured concrete increased by 18.27%, 56.50%, and 75.15%, while the corresponding values of the naturally cured concrete increased by 20.95%, 39.20%, and 57.33%, respectively, at the 7th, 28th, and 90th day.

From the above test results, steam curing significantly improved the early compressive strength of the concrete, while the compressive strength of the steam-cured concrete was lower than that of the standard-cured concrete after 28 days. The reasons for the compressive strength changes were as follows: During steam curing, the hydrated products of the hydrated calcium silicate gel grew denser, and they were unable to effectively fill the pores, which decreased the porosity and strength of the hardened concrete. The naturally cured concrete presented the lowest value for each aging condition due to its poor curing conditions, lower temperature, and humidity compared to the other two curing conditions.

For the splitting tensile strength shown in Figure 14, the trend of the splitting tensile strength development was similar to that of the compressive strength. Compared to the splitting tensile strength of the cured concrete at the 3rd day, the splitting tensile strength of the steam-cured concrete increased by 4.85%, 7.52%, and 9.95% at the 7th, 28th, and 90th day, respectively. The corresponding values of the standard-cured concrete were 25.35%, 65.49%, and 83.80%, and the corresponding values of the naturally cured concrete were 22.04%, 53.06%, and 69.80%.

Based on the above experimental results, the effects of the curing conditions on the concrete strength were comprehensively studied. It was concluded that in the initial stage, the steam curing condition significantly improved the strength of the concrete, followed by the standard curing condition, and the natural curing condition presented the weakest performance. At the 28th day, the strength of the standard-cured concrete was higher than that of the steam-cured concrete. After 28 days, the rate of the increase in the strength of the standard-cured concrete was higher than that of the steam-cured concrete. As the age increased, the increasing rate of the splitting tensile strength was lower than that of the compressive strength, indicating that the brittleness of the concrete increased as the age increased.

### 3.4. The Fracture Property of Concrete

The methods for calculating fracture toughness $K_{\mathrm{IC}}^{S}$, critical crack tip opening displacement ${\mathrm{CTOD}}_{c}$, and critical strain energy release rate $G_{\mathrm{IC}}^{S}$ are given in [24,25].

(1) The fracture toughness $K_{\mathrm{IC}}^{S}$ of the concrete under the three curing conditions is shown in Table 8 and Figure 15. As the age increased, the degree of hydration increased, and thus, $K_{\mathrm{IC}}^{S}$ increased. Therefore, the crack propagation resistance of the concrete improved.

Steam curing can significantly improve the early fracture toughness of concrete. After 28 days, the standard curing condition was better than the steam curing in improving the fracture toughness $K_{\mathrm{IC}}^{S}$ of the concrete. Standard curing was also more conducive for the development of long-term crack resistance of concrete.

(2) The critical crack tip opening displacements ${\mathrm{CTOD}}_{c}$ of the concrete under the three curing conditions are shown in Table 9 and Figure 16. The results showed that the ${\mathrm{CTOD}}_{c}$ increased as the age increased. A sharp increase in the ${\mathrm{CTOD}}_{c}$ mainly occurred before the 3rd day, and ${\mathrm{CTOD}}_{c}$ development was stable after 3 days.

The effect of the curing condition on the ${\mathrm{CTOD}}_{c}$ was found in its growth rate before the 28th day. Before the 7th day, the ${\mathrm{CTOD}}_{c}$ of the steam-cured concrete was the largest. However, after the 28th day, the ${\mathrm{CTOD}}_{c}$ of the standard-cured concrete was the largest, and the trend was similar to that of $K_{\mathrm{IC}}^{S}$.

Based on ${\mathrm{CTOD}}_{c}$ development, when a concrete crack was about to develop, the critical value of the allowable crack opening for the standard-cured concrete was larger than that of the steam-cured concrete. Furthermore, the allowable deformation of the steam-cured concrete was smaller. Therefore, steam curing significantly increases the brittleness of concrete.

(3) The critical strain energy release rate $G_{\mathrm{IC}}^{S}$ of the cured concrete samples are shown in Table 10 and in Figure 17. As the time increased, the hydration reaction degree increased, and $G_{\mathrm{IC}}^{S}$ increased. The energy per unit length of crack propagation increased, and the ability of the concrete to resist crack propagation improved.

The results showed that the steam curing significantly increased the growth rate of $G_{\mathrm{IC}}^{S}$. On and before the 7th day, the$\;G_{\mathrm{IC}}^{S}$ of the steam-cured concrete was the largest. From the 28th day and onwards, the $G_{\mathrm{IC}}^{S}$ of the standard-cured concrete was the largest. From the 28th day, $G_{\mathrm{IC}}^{S}$ of all the cured concrete developed similarly. The development of the $G_{\mathrm{IC}}^{S}$ of the cured concrete indicated that the standard curing condition could improve crack propagation resistance better than steam curing.

### 3.5. The Brittleness of Concrete

(1)Material length Q and characteristic length $l_{\mathrm{ch}}$

The characteristic length $l_{\mathrm{ch}}$ and material length *Q* were applied as the criteria for concrete brittleness, and the relevant theories are given in [26]. The material length Q and characteristic length $l_{\mathrm{ch}}$ of the cured concrete are shown in Table 11 and Figure 18. Q and $l_{\mathrm{ch}}$ showed downward trends as the age increased. On the 7th day and before, the Q and $l_{\mathrm{ch}}$ of the steam-cured concrete were the largest. On the 28th day and later, the Q and $l_{\mathrm{ch}}$ of the standard-cured concrete were the largest, while the Q and $l_{\mathrm{ch}}$ of the naturally cured concrete were the smallest for each age. Compared to the standard curing condition, the rates at which Q and $l_{\mathrm{ch}}$ decreased were higher, indicating that steam curing increases the brittleness of concrete.

(2)Tension–compression strength ratio

The tension–compression strength ratios of the cured concrete are shown in Table 12 and in Figure 19. From the 28th day, the tension–compression strength ratio decreased as time increased, indicating that the brittleness of the cured concrete increased. By the 90th day, the tension–compression strength ratio of the steam-cured concrete decreased by 0.70% compared to on the 28th day. The corresponding ratios for the standard and naturally cured concrete were 0.80% and 1.86%, respectively.

Based on the tension–compression strength ratio data before the 28th day, a clear development trend could not be obtained. Therefore, the tension–compression strength ratio was suitable for evaluating the brittleness development from a long-term perspective, but not for that at an early age.

## 4. Discussion

### 4.1. Effect of Curing Condition on Fracture Toughness $K_{\mathrm{IC}}^{S}$

In the steam-cured concrete, the thermal effect weakened the transition zone in the concrete and caused fracture toughness $K_{\mathrm{IC}}^{S}$ development. The transition zone was the weakest link in the internal structure and can be regarded as the limiting factor for concrete strength. The low strength of the transition zone was mainly due to the presence of microcracks. High-temperature curing resulted in thermal cracks, which negatively impacted the compactness of the concrete transition zone. Thus, during steam curing, the cooling stage was the main reason for the damage that was observed in the concrete transition zone. When the temperature dropped, the surface layer of the concrete cooled faster and shrank faster than the inner layer. When there is a large temperature difference between the surface and concrete core, tensile stress builds up on the surface and causes crack formation. From a micro perspective, due to the temperature difference, and the differences in the thermal expansion coefficients of the aggregate and cement paste, the displacement between the aggregate and cement paste in the transition zone was different. Therefore, this resulted in stress concentrations or excessive tensile stresses and cracking in the transition zone, reducing the ability of the concrete to resist crack propagation.

### 4.2. Effect of Curing Condition on Critical Crack Tip Opening Displacement ${\mathrm{CTOD}}_{c}$

In the steam curing process, the hydration reaction was intensified during the initial stage. A thick hydrate-shielding film formed on the surface of the particles, immediately leading to a decreased hydration rate, and the shielding film became denser with further hydration. The density of the gel increased by 15–20% compared to that at normal temperatures. This indicated that the shielding film could not be permeated easily and that it blocked the internal diffusion of the hydration system inside the concrete, which affected the reaction rate and extent of the concrete during the later hardening stage.

Under the steam curing condition, the size of the hydrated crystals increased as the curing temperature increased, forming coarse-grained structures, and the specific surface area of the hydrated crystal decreased, resulting in a decrease in dispersion. The large crystals adversely affected the strength and toughness of the concrete structure. As the temperature decreased, the specific surface area and unit volume concentration of the hydrate increased, and the contact points between the particles increased. The binding forces between the particles were mainly determined by van der Waals interactions and electrostatic forces. The increase in the number of contact points between the particles resulted in an increase in cohesive forces. Therefore, the critical crack tip-opening displacement of the steam-cured concrete was less than the corresponding value of the standard-cured concrete in the later stages.

### 4.3. Effect of Curing Condition on Critical Strain Energy Release Rate $G_{\mathrm{IC}}^{S}$

The reason for the decrease in the $G_{\mathrm{IC}}^{S}$ growth rate of the steam-cured concrete may be due to the large number of hydration products that were produced during the early stage of curing. The damage in the internal structure of the concrete occurred due to high temperatures. The expansion coefficient of the hydrated calcium silicate gel and coarse aggregate was different, and potential cracks in the transition zone formed under the temperature gradient. Additionally, a large number of gel deposits created micro cracks in the concrete.

Standard curing improved the $G_{\mathrm{IC}}^{S}$ growth rate because the standard curing condition produced a more compact concrete internal structure, avoiding damage caused by the excessive temperature gradient inside the concrete.

### 4.4. Long-Term Performance Predication of Concrete

When loaded at an early age, a damage zone occurs in the concrete, and the damage zone will evolve into a critical state, followed by the creation of macro cracks. The damage and fracture of a concrete structure is a continuous process, and the energy release rate is the link between damage and fracture. In the critical state, the relationship between concrete damage and fracture can be established by their common relationship to the energy release. The previous experimental results reveal the effect of the loading age and the curing condition on the strength, fracture property, and brittleness development of concrete. Being able to accurately predict the long-term performance of concrete under the coupled effects of loading and curing is of great significance. Therefore, it is necessary to establish a time-dependent model of concrete fracture toughness under the coupled effect of loading and curing.

## 5. The Time-Dependent Model of Concrete Fracture Toughness

A time-dependent model of concrete fracture toughness that considers the curing conditions and the loading ages is proposed to predict the fracture properties of concrete. The model’s derivation process is described in this section.

### 5.1. Model Establishment

#### 5.1.1. Relationship between Fracture Toughness and Crack Propagation Resistance

The fracture criterion for crack instability propagation is $G\geq G_{\mathrm{IC}}^{S}$. From the stress field near the crack tip, the fracture toughness $K_{\mathrm{IC}}^{S}$ in the two-parameter fracture model is obtained. Using the mode I flat plate crack as an example, the relationship between stress intensity factor $K_{I}$ and energy release rate $G_{I}$ is given by Equation (1):(1)$$G_{I}=\frac{K_{I}{}^{2}}{E_{I}},$$

In the critical state, the relationship is given by Equation (2):(2)$$G_{IC}=\frac{K_{IC}{}^{2}}{E},$$

In fracture mechanics, the criterion for fracture is $G\geq G_{\mathrm{IC}}$. Therefore, the energy release rate G on the left side can be used to measure whether a crack will propagate, and this is the crack propagation force. $G_{\mathrm{IC}}$ on the right side denotes the resistance of crack propagation, which reflects the ability of the material to prevent cracks, as expressed by R. When the crack propagation force is greater than the crack propagation resistance, the crack begins to propagate. In this case, the energy release rate G is shown in Equation (3):(3)$$G=\frac{\pi \sigma^{2}a}{E_{I}},$$

Based on Equation (3), Figure 20 shows the relationship of R, G, and a. The horizontal axis is the crack length a, and the vertical axis is the crack propagation resistance R and crack propagation force G. When the tensile stress is constant, the crack propagation force G increases linearly as a increased. When brittle fracture occurs, the crack propagation resistance R is determined by $G_{\mathrm{IC}}$, and the fracture toughness is a material constant that does not change with the crack length, a. Therefore, the dotted line represents the crack propagation resistance line.

When the tensile stress $\sigma_{1}$ is applied, the crack propagation resistance R is reached when the length of the crack reaches $a_{1}$, and instability and fracture occur when the length of the crack exceeds $a_{1}$, and this is also true when tensile stress $\sigma_{2}$ is applied. Thus, Equation (4) was obtained:(4)$$R=\frac{K_{IC}{}^{2}}{E},$$

#### 5.1.2. Study on Mode I Crack Damage and Fracture Based on Energy Release

(1) The crack-tip damage zone in concrete is the transition zone at the crack tip, which is a micro-crack zone that is caused by tensile stresses. It is appropriate to use the maximum tensile stress theory to study the mode I crack of concrete.

Based on the maximum tensile stress theory:(5)$$\sigma_{1}=\sigma_{u},$$
where $\sigma_{1}$ is the first principal stress in elasticity and the maximum tensile stress; $\sigma_{u}$ is the tensile strength of the material.

(2) The damage strain energy release rate |y| can be calculated according to the following equations:

For damaged materials, the elastic strain energy is given by:(6)$$\rho \phi_{e}\left(\varepsilon^{e},D\right)=\frac{1}{2}\varepsilon^{e}:E\left(D\right):\varepsilon^{e},$$

The elastic complementary energy is given by:(7)$$\rho \psi_{e}\left(\sigma ,D\right)=\frac{1}{2}\sigma :E^{-1}\left(D\right):\sigma ,$$

The damage strain energy release rate is:(8)$$y=-\frac{\partial \psi_{e}}{\partial D}=\frac{\partial \phi_{e}}{\partial D}=-\frac{1}{2}\varepsilon^{e}:\frac{\partial E\left(D\right)}{\partial D}:\varepsilon^{e},$$

Equation (9) is obtained after tensor calculus:(9)$$\left|y\right|=\frac{1+\nu }{2E{\left(1-D\right)}^{2}}\sigma^{2}-\frac{\nu }{2E{\left(1-D\right)}^{2}}{\left(t_{t}\sigma \right)}^{2},$$

Therefore, the hydrostatic pressure can be determined by $\sigma_{H}=\frac{1}{3}t_{r}\left(\sigma \right)$, and the deviatoric stress tensor as $\sigma^{\prime }=\sigma -\sigma_{H}I$, where the von Mises equivalent stress is given by Equation (10):(10)$$\sigma_{eq}={\left[\frac{3}{2}\left(\sigma -\sigma_{H}I\right):\left(\sigma :\sigma_{H}I\right)\right]}^{\frac{1}{2}},$$

Equation (11) can be obtained from Equation (9):(11)$$\left|y\right|=\frac{\sigma_{\mathrm{eq}}^{2}}{2E{\left(1-D\right)}^{2}}s_{t},$$
where $s_{t}$ is the triaxial stress factor, which is expressed by Equation (12):(12)$$s_{t}=\frac{2}{3}\left(1+\upsilon \right)+3\left(1-2\upsilon \right){\left(\frac{\sigma_{H}}{\sigma_{eq}}\right)}^{2},$$

Therefore, a mode I crack under uniaxial stress can be expressed, where $\sigma_{H}=\frac{1}{3}\sigma$, $\sigma_{eq}=\sigma$, $s_{t}=1$:(13)$$\left|y\right|=\frac{\sigma^{2}}{2E{\left(1-D\right)}^{2}},$$

Where Equation (13)’s equivalent stress is $\sigma_{eq}={\left[\frac{3}{2}\left(\sigma -\sigma_{H}I\right):\left(\sigma -\sigma_{H}I\right)\right]}^{\frac{1}{2}}$.

(3) The following is a discussion on the relationship between G and |y|. When the concrete bears the load, a damaged area appears first. When the load increases, the damaged area increases, resulting in a release of energy. When the critical state is reached, the fracture zone appears, and a critical value for energy release also appears, with the damage energy release rate being equal to the fracture energy release rate. When the load increases further, cracks appear in the concrete, and the cracks expand as the load increases.

The cracks are generated in the damage zone and are accompanied by the release of energy. When the material is damaged by loading, the damage deterioration process will be shortened, accelerating the fracture of the material.

Equation (14) presents the energy relationship in the critical state:(14)$$G_{I}=\underset{a\to 0}{lim}\frac{\Delta \phi_{e}}{\Delta a}=\underset{D\to D_{C}}{lim}\frac{\Delta \phi_{e}}{\Delta D}=\left|y\right|,$$
where $\phi_{e}$ is the elastic strain energy; D is the damage variable; a is the crack length; and $D_{C}$ is the critical damage variable.

Equation (14) expresses that G and |y| are constant at the beginning of macroscopic crack formation, and they are in a critical state. Equation (14) also establishes the relationship between damage and fracture. However, this study only focuses on the mode I crack, which is an open crack, and the loading mode is the same as that of the Loland model. Therefore, the Loland model is used to study the coupling relationship between damage and fracture.

(4) The damage and fracture coupling of the mode I crack are analyzed in this section. The tip of the mode I crack exists as a microcrack damage zone, the shape and size of this zone are depicted in Figure 21 and Equation (15):
(15)$$r\left(\theta \right)=\frac{K_{I}^{2}}{2\pi \sigma_{u}^{2}}{\left[\cos\frac{\theta }{2}\left(1+\sin\frac{\theta }{2}\right)\right]}^{2},$$
where $\sigma_{u}$ is the tensile strength; $K_{I}$ is the stress intensity factor; and r is the radius of the microcrack damage zone.

For a mode I crack, the expression of K is given in Equation (16):(16)$$K=\sigma \sqrt{\pi a}f\left(\frac{a}{h}\right).$$

(5) The equivalent principle of effective stress and strain can also be obtained. In damage mechanics, the damage variable is defined as the ratio of the damaged area $\tilde{A}$ to an area without damage A, $D=\tilde{A}/A$. Effective stress $\tilde{\sigma }$ is defined by Equation (17):(17)$$\tilde{\sigma }=\frac{F}{A-\tilde{A}}=\frac{F/A}{1-\tilde{A}/A}=\frac{\sigma }{1-D},$$
where σ is the nominal stress.

Without considering the material damage, the elastic constitutive equation without considering material damage is $\varepsilon =\sigma /E_{0}$. The nominal stress is replaced by the effective stress, and the elastic constitutive equation of the damaged material can be obtained by Equation (18):(18)$$\varepsilon =\frac{\tilde{\sigma }}{E_{0}}=\frac{\tilde{\sigma }}{E_{0}\left(1-D\right)},$$

The stress–strain relationship of the damaged concrete is given by Equation (19):(19)$$\sigma =E_{0}\left(1-D\right)\varepsilon ,$$

The expression of the mode I crack stress intensity factor of concrete considering damage is:(20)$$K_{I}=E_{0}\left(1-D\right)\varepsilon \sqrt{\pi a},$$

The effective stress of the materials can be expressed by Equation (21):(21)$$\tilde{\sigma }=\frac{\sigma }{\left(1-D\right)},$$

Additionally, the constitutive relation follows:(22)$$\tilde{\sigma }=\{\begin{array}{c} \tilde{Ε}\varepsilon \;\;\;\;\left(0\leq \varepsilon \leq \varepsilon_{c}\right)\\ \tilde{E}\varepsilon_{c}\;\;\left(\varepsilon_{c}\leq \varepsilon \leq \varepsilon_{u}\right) \end{array},$$
where $\varepsilon_{u}$ is the ultimate strain when D=1; $\varepsilon_{c}$ is the peak strain; and $\tilde{Ε}$ is the net elastic modulus of damaged concrete, according to:(23)$$\tilde{Ε}=\frac{E}{1-D_{0}},$$
where E is the elastic modulus of the undamaged material, and $D_{0}$ is the initial damage to a material before loading.

#### 5.1.3. Effect of Loading Age on Fracture Toughness

(1) To assess the damage caused by immediate strain, the Loland damage model considers the initial damage of materials and considers that the damage will occur when the load is applied, which indicates that the damage evolves gradually under load.

The data obtained from the uniaxial tensile test of concrete were fitted, and the damage evolution is given by Equation (24):(24)$$D_{L}=\{\begin{array}{c} D_{0}+C_{1}\varepsilon_{e}{}^{\beta }\;\;\;\;\;\;\;\;\;\;\;\;\;\;\;\;\;\;\;\;\;\;\;\;\;\left(0\leq \varepsilon \leq \varepsilon_{c}\right)\\ D_{0}+C_{1}\varepsilon_{c}^{\beta }+C_{2}\left(\varepsilon_{e}-\varepsilon_{c}\right)\;\;\;\;\;\left(\varepsilon_{c}\leq \varepsilon \leq \varepsilon_{u}\right) \end{array},$$
where $C_{1},$ $C_{2}$, and β are the material constants, and when $\varepsilon =\varepsilon_{c}$, $\sigma =\sigma_{c}$, $\frac{d\sigma }{d\varepsilon }=0$, and when $\varepsilon =\varepsilon_{c}$, D=1. Therefore, Equation (25) can be obtained:(25)$$\beta =\frac{\lambda }{1-D_{0}-\lambda },C_{1}=\frac{\left(1-D_{0}\right)\varepsilon_{c}^{-\beta }}{1+\beta },C_{2}=\frac{1-D_{0}-C_{1}\varepsilon_{c}^{\beta }}{\varepsilon_{u}-\varepsilon_{c}},$$
where $\lambda =\sigma_{c}/(\tilde{E}\varepsilon_{c}$).

According to [27], the critical damage variable $D_{c}=0.18$, initial damage $D_{0}=0.05$, peak stress $\sigma_{c}=2.5\;\mathrm{MPa}$, peak strain $\varepsilon_{c}=10^{-4}$, ultimate strain $\varepsilon_{u}=5\times 10^{-4}$, initial elastic modulus$\;E_{0}=3\times 10^{4}\;$MPa, and tensile strength $\sigma_{u}=3.0\;$MPa.

The above data are substituted into Equation (23); therefore, $\tilde{Ε}=3.16\times 10^{4}$ MPa, and λ=0.79, β=5, $C_{1}=1.58\times 10^{19}$, and $C_{2}=376$ are obtained. The material constants mentioned above are substituted into the damage evolution equation, and Equation (26) is obtained:(26)$$D_{L}=\{\begin{array}{c} 0.05+1.58\times 10^{19}\times \varepsilon_{e}{}^{5}\;\left(0\leq \varepsilon \leq 10^{-4}\right)\\ 0.208+376\times \left(\varepsilon_{e}-\varepsilon_{c}\right)\;(10^{-4}<\varepsilon \leq \varepsilon_{u}) \end{array}.$$

(2) The creep deformation caused by the load increases the internal strain energy of the material. When the accumulated strain energy exceeds a certain limit, some strain energy is released due to the micro-crack and will be converted into the surface energy of the crack, which causes crack initiation and expansion inside the concrete. The overall elastic modulus of the concrete decreases with the development of microcracks; thus, creep deformation increases. The coupling effect of damage and creep causes the cracks to develop more rapidly, leading to concrete failure. Based on the statistical damage constitutive model and the randomness of defect distribution, the internal damage $D_{y}$ of the concrete under long-term continuous load is considered.

The damage variable $D_{y}$ is defined as the ratio of the number of damaged micro units $N_{F}$ to the total number of units N in the material. Combined with the continuous damage statistical strength theory, it can accurately reflect the mechanical properties of brittle materials.

Assuming that P obeys the Weibull distribution, its probability density distribution function can be expressed by Equation (27):(27)$$P\left(F\right)=\frac{m}{F_{0}}{\left(\frac{F}{F_{0}}\right)}^{m-1}\exp\left[-{\left(\frac{F}{F_{0}}\right)}^{m}\right],$$
where P is the unit strength of the concrete material; m and $F_{0}$ are the concrete Weibull parameters; and F is the Weibull distribution variable of the micro-unit. When F exceeds $F_{0}$, the micro-unit begins to damage, according to:(28)$$N_{F}=\int_{0}^{N}NP\left(x\right)dx,$$

Equation (27) is substituted into Equation (28), and the damage variable $D_{y}$ can be expressed by Equation (29):(29)$$D_{y}=\int_{0}^{F}P\left(x\right)dx=1-\exp\left[-{\left(\frac{F}{F_{0}}\right)}^{m}\right],$$

Based on Equation (29), determining the micro-unit strength F is the most important value in the statistical damage model. F is defined as the effective strain energy of the micro-unit, where $F=\sigma \cdot \varepsilon_{c}$, σ is the effective stress, and $\varepsilon_{c}$ is the effective strain. The failure condition of the micro-unit occurs when the strain energy exceeds the critical value.

Based on the test, m=3.51 and $F_{0}=4.85\times 10^{-4}$ MPa. The damage variable $D_{y}$ is given by Equation (30):(30)$$D_{y}=\int_{0}^{F}P\left(x\right)dx=1-\exp\left\{-[\frac{\left(\sigma \cdot \varepsilon_{cc}\right)}{\left(4.85\times 10^{-4}\right)}{]}^{3.51}\right\}.$$

(3) Tensile creep $\varepsilon_{\mathrm{cc}}$ is assessed. The MC2010 model is suitable for ordinary concrete, where compressive stress < 0.4$f_{ck}$, the average compressive strength is 20–130 MPa after 28 days, the relative humidity is 40–100%, average temperature is 5–30 °C, and the loading age > 1 day. Therefore, this model was suitable for assessing tensile concrete, and the MC2010 model [28] was applied in this study.

The creep of concrete under $\sigma \left(t_{0}\right)$ is given by Equation (31):(31)$$\varepsilon_{cc}\left(t,t_{0}\right)=\frac{\sigma \left(t_{0}\right)}{E_{ci}}\varphi \left(t,t_{0}\right),$$
where $\varepsilon_{cc}\left(t,t_{0}\right)$ is the creep of concrete, and $E_{ci}$ is the elastic modulus of concrete after the 28th day.

For the compressive strength of concrete at the 28th day $f_{cm}$, Equation (32) follows:(32)$$E_{ci}=E_{c0}\cdot \alpha_{E}\cdot {\left(\frac{f_{cm}}{10}\right)}^{\frac{1}{3}},$$
where $E_{c0}=21.5\times 10^{3}$ MPa, and $\alpha_{E}$ is the coefficient related to aggregate type, which in this case, consisted of limestone, $\alpha_{E}=0.9$.

The creep coefficient of concrete is given by Equation (33):(33)$$\varphi \left(t,t_{0}\right)=\varphi_{0}\beta_{c}\left(t,t_{0}\right)=\varphi_{H}\beta \left(f_{cm}\right)\beta \left(t_{0}\right)\beta_{c}\left(t,t_{0}\right),$$
where $\varphi_{0}$ is the nominal creep coefficient; $\beta_{c}\left(t,t_{0}\right)$ is time function of the creep process; $\varphi_{H}$ is the relative humidity correction factor; $\beta \left(f_{cm}\right)$ is the correction factor of the concrete strength; and $\beta \left(t_{0}\right)$ is the correction factor of the loading age of the concrete, according to:(34)$$\varphi_{H}=\left[1+\frac{1-RH/100}{0.1\cdot \sqrt[{3}]{h}}\cdot \alpha_{1}\right]\cdot \alpha_{2},$$
(35)$$\beta \left(f_{cm}\right)=16.8/\sqrt{f_{cm}},$$
(36)$$\beta \left(t_{0}\right)=\frac{1}{0.1+{\left(t_{0}\right)}^{0.2}},$$
(37)$$\beta_{c}\left(t,t_{0}\right)={\left[\frac{t-t_{0}}{\beta_{H}+\left(t-t_{0}\right)}\right]}^{0.3},$$
(38)$$\beta_{H}=1.5\cdot h\cdot [1+\left(1.2\cdot RH/100{)}^{18}\right]+250\alpha_{3}\leq 1500\alpha_{3},$$
(39)$$\alpha_{1}=[\frac{35}{f_{cm}}{]}^{0.7},\alpha_{2}=[\frac{35}{f_{cm}}{]}^{0.2},\alpha_{3}={\left[\frac{35}{f_{cm}}\right]}^{0.5},$$
where RH is the relative humidity of the surrounding environment; h is the effective thickness of the section; $h=2A_{C}/u$, $A_{C}$ is the cross-section area of the section; u is the circumference of the specimen in contact with the atmosphere; t is concrete age; $t_{0}$ is the loading age of the concrete; $f_{cm}$ is the average compressive strength at the 28th day; and $f_{cm}=f_{ck}+\Delta f$, Δf=8 MPa, $f_{ck}$ is the characteristic value of concrete strength, and $f_{ck}=0.8f_{cu,k}$, $\sigma \left(t_{0}\right)=1.959\;$MPa, RH = 40%, h = 50 mm, $f_{cu,k}$ = 50 MPa.

(4) The immediate strain of concrete $\varepsilon_{e}$ is given by:(40)$$\varepsilon_{e}\left(t_{0}\right)=\frac{\sigma \left(t_{0}\right)}{E_{ci}\left(t_{0}\right)},$$
where $\varepsilon_{e}\left(t_{0}\right)$ is the immediate strain of the concrete, and $E_{ci}\left(t_{0}\right)$ is the elastic modulus at loading age $t_{0}$; therefore,
(41)$$E_{ci}\left(t_{0}\right)=\beta_{E}\left(t_{0}\right)E_{ci},$$
where $\beta_{E}\left(t\right)$ is the coefficient related to concrete, which is given by:(42)$$\beta_{E}\left(t_{0}\right)={\left[\beta_{cc}\left(t_{0}\right)\right]}^{0.5},$$
where $\beta_{cc}\left(t\right)$ is the function that describes time development, which is given by:(43)$$\beta_{cc}\left(t_{0}\right)=\exp\{s\cdot [1-\left(\frac{28}{t_{0}}{)}^{0.5}\right]\},$$
where s is the coefficient depending on the strength grade of the cement, and for 42.5R cement, s=0.2.

#### 5.1.4. Effect of Curing Conditions on Fracture Toughness

The coupling effect of temperature and time can be considered according to maturity, and the relationship between fracture parameters and maturity can be established.

The previous experimental data and discussion indicated that the fracture toughness was approximately logarithmic with the age, and this regulation was consistent with the research results that have been obtained by many scholars. For example, the research conducted by Mi Zhengxiang [29] showed that the fracture behavior of concrete was approximately logarithmic with the age. Furthermore, Lin Chen [30] believed that the fracture toughness of concrete increased predictably with an increase in compressive strength, and the relationship between fracture toughness and compressive strength was approximately in the form of a double broken line, and the turning point was 40% of the compressive strength of concrete at the 28th day. In addition, $K_{IC}$ changed logarithmically with the age because the setting degree and hydration degree at each age were different. Therefore, in the early stages of cement hydration, $K_{IC}$ increased more intensely, and then gradually slowed down with the age.

In this study, maturity was introduced to include the effects of curing condition, temperature, and age on fracture characteristics. Combined with the fracture test results of the concrete under different curing conditions at each age, the relationship between fracture parameters and maturity of the concrete was studied. Finally, a relationship between fracture toughness and the equivalent maturity of the concrete was obtained.

Nures [31] and Saul [32] defined the maturity M of concrete as a function of temperature T and time t:(44)$$M=\sum_{0}^{t}\left(T-T_{0}\right)△t,$$
where M is maturity, °C×d; T is the average temperature in Δt, °C; $T_{0}$ is the reference temperature, °C; where $T_{0}$ = −10 °C; *t* is the total time, d; and Δt is the time step, d.

The concrete maturity was calculated by Equation (44), and the development model of the fracture toughness for concrete maturity was established in [33]. However, the correlation coefficient $R^{2}$ was only above 0.56.

Based on the double-broken line relationship between fracture toughness and compressive strength from [34], the effect of the curing temperature on the fracture properties of concrete before 28 days had passed was greater than the mechanical properties. However, after a certain age, the effect decreased, and the fracture parameters increased with the age. This may explain the low correlation coefficient. Considering the effects of temperature and age on the fracture parameters of the concrete, the proportion of temperature effect in the first 28 days should be increased. Therefore, the equivalent maturity proposed by Guan Junfeng [33] was applied:(45)$$M_{e}=\sum_{0}^{t}T_{e}△t,$$
where $M_{e}$ is equivalent maturity, °C×d; $T_{e}$ is the equivalent temperature, °C; t is the age, d; and Δt is curing time, d.

In Equation (44), the temperature is the internal temperature of the concrete. Because the specimen was small, the internal temperature of the concrete could disperse quickly, and the increase of temperature was not high. Ni Tongyuan [34] found that the internal temperature increase in concrete was less than 5 °C before 0.8 days; thus, the effect of the internal temperature in concrete was not considered. Due to the different effect proportions of temperature and age, the curing temperature was used as the equivalent temperature before the 28th day, and the average curing temperature during the first 28 days was used as the equivalent temperature after the 28th day.

For three days of steam curing, the temperature was 45 °C, and natural curing was applied from the 3rd to the 28th day, the temperature was 45 °C, and the equivalent temperature was $T_{e}=17.77\;{}^{\circ}C$. For standard curing for 28 days, the temperature was 21 °C, and the equivalent temperature was $T_{e}=21\;{}^{\circ}C$. In the samples that had been naturally cured for 28 days, the temperature was 14.5 °C, and the equivalent temperature was $T_{e}=14.5\;{}^{\circ}C$. After the 28th day, the equivalent maturity of the three curing methods was as follows:(46)$$\mathrm{steam}\;\mathrm{curing}:\;M_{e}=17.77t,$$
(47)$$\mathrm{standard}\;\mathrm{curing}:\;M_{e}=21t,$$
(48)$$\mathrm{natural}\;\mathrm{curing}:\;M_{e}=14.5t.$$

By substituting the equivalent maturity into the relationship between the maturity and stress intensity factors, the change in the stress intensity factor with time under different curing conditions could be obtained:(49)$$K_{IC}=\alpha \cdot ln({Μ}_{e})-\beta ,$$

The time-dependent model of concrete fracture toughness under different curing conditions and continuous load could therefore be expressed by:(50)$$\left\{ \begin{array}{l} |R=\frac{K_{IC}^{2}}{E}=\frac{{\left[(1-D_{L})(1-D_{y})K_{IC}\right]}^{2}}{E}\\ |D_{L}=0.05+1.58\times 10^{19}\varepsilon_{e}^{5}\\ |D_{y}=1-\exp[-(\frac{\left(\sigma 0\cdot \varepsilon_{cc}\right)}{4.85\times 10^{-4}}{)}^{3.51}],\\ |K_{IC}=\alpha \cdot ln(T_{e}\cdot t)-\beta \\ |\varepsilon_{cc}\left(t,t_{0}\right)=\frac{\sigma \left(t_{0}\right)}{E_{ci}}\varphi \left(t,t_{0}\right),\\ |E=\beta_{E}\left(t_{0}\right)E_{ci} \end{array}\right.$$

In Equation (50), $T_{e}$, α, and β are the pending parameters, and s, RH, h, $\alpha_{E}$, $f_{cu,k}$, and ${\sigma (t}_{0})$ are the unknown parameters in the creep equation, and $t_{0}$ is the age when load begins to apply.

### 5.2. Parameter Calibration

Some parameters in Equation (50) need to be determined, and s, RH, h, $\alpha_{E}$, and $f_{cu,k}$ can be obtained from the composition of concrete and the related material parameters; $T_{e}$ is obtained from the curing condition, ${\sigma (t}_{0})$ and $t_{0}$ are controlled by the load value and the age, and α and β are unknown and can be determined by regression analysis according to the test data.

The experimental value of the crack propagation resistance R (critical strain energy release rate $G_{\mathrm{IC}}^{S}$) of the steam-cured concrete loaded at the 3rd day was selected for regression analysis. The test results are shown in Table 13, and the relationship between R and age is shown in Figure 22. The known parameters are shown in Table 14.

Based on Table 13 and Table 14, Equation (50) is fitted by least square regression. The corresponding fitting results are shown in Table 15, and the fitting curve is shown in Figure 23.

From Table 15, for the steam-cured concrete loaded at the 3rd day, the coefficient of determination for the correction fitted by Equation (50) is higher than 0.99, and the curve trend is close to the measured R value.

All of the unknown parameters in the time-dependent model of concrete fracture toughness are obtained. The time-dependent model of concrete fracture toughness under different curing conditions and loading ages can thus be described by substituting the parameter values into Equation (50).

### 5.3. Verification

Equation (50) was verified by the experimental results of crack propagation resistance of the cured concrete under different curing conditions. The measured R values of the standard cured concrete were selected as the criterion. The error comparison between the predicted values and the measured values from Table 10 are shown in Table 16, and the relationship between the model curve and the measured values is described in Figure 24.

As observed in Table 16 and Figure 24, the differences between the predicted values of the model and the measured values are small for each age, and the maximum error rate is 5.65%. The analysis shows that this new model exhibits high accuracy and is more consistent with the actual condition. It also verified that the mechanism explanations that are provided in this study are reasonable.

In addition, the fracture toughness $K_{\mathrm{IC}}^{S}$ data from [30,35] were selected as reference data for further verification and analysis. The comparison between the predicted value and the values in [30,35] are shown in Table 17 and Table 18. In Table 17 and Table 18, the model-predicted values should be compared to the test values that were obtained in [30,35] due to the difference in experimental the conditions and the complexity of the concrete’s internal structure, where an error rate of about 10% is acceptable, indicating that the model presents strong practicality. In the unknown parameter fitting of the proposed model, the data samples come from the test results of beam specimens at 3 days, 7 days, 28 days, and 90 days under 3 curing conditions and at 3 loading ages.

In Table 17, based on the parameters of the concrete beams in [30], the errors between the predicted value and the test values at 1 day and 2 days are 10% and 16%, respectively. The reason for this could be that the equivalent maturity of the model reflecting the hydration process is based on the fracture toughness data of the specimen after 3 days, which is not suitable for evaluating the development of fracture toughness before 3 days. In addition, in practical engineering, precast beams are being cured and are under no-load within the first 3 days, for which the importance of predicting fracture toughness development is lower than that of the later stage. At the age of 7 and 28 days, the error between the model predicted value and the measured value of the model is less than 6%, indicating that the model can accurately predict the long-term development of fracture toughness.

In Table 18, based on the parameters of the concrete beams in [35], the errors between the model-predicted values and the test values in the curing conditions of 5 °C and 60 °C are 9.4% and 12.57%, respectively. The reason for this be that the model is based on the expressions of equivalent temperatures at 17.77 °C, 21 °C, and 14.5 °C, which correspond to steam, standard, and natural curing. The development of fracture toughness cannot be accurately predicted for a low curing temperature of 5 °C and a high curing temperature of 60 °C. At 20 °C and 40 °C, the error between the model-predicted values and the measured values is within 6.3%, indicating that the model can accurately predict the development of fracture toughness in normal curing temperatures.

In conclusion, the model can accurately predict the development of concrete fracture toughness.

## 6. Conclusions

In this study, the effect of the curing condition and loading age on the mechanical, fracture, and brittleness properties of concrete was investigated at four different ages (3, 7, 28, and 90 days). Steam, standard, and natural curing conditions and loading ages of 3, 7, and 28 days were considered. A time-dependent model of concrete fracture toughness that considered the effects of the curing condition and the loading age was established. The key parameters were calibrated based on the test data, and the suitability of the model was investigated. The conclusions can be summarized as follows:

(1) Based on the P-CMOD curves of the concrete samples that were created under different curing conditions, steam curing improved the early strength of concrete but weakened subsequent strength in the later stages. Based on the crack initiation load and load limit of the concrete, the steam curing condition significantly improved the early bearing capacity of the concrete and had adverse effects on its later strength. The standard curing condition had a better effect on the later bearing capacity of the concrete, and the natural curing condition had a weaker effect on the early and later bearing capacity of the concrete compared to the abovementioned two curing methods. Based on the results of the strength tests, it was shown that steam curing significantly improved the early compressive strength and the splitting tensile strength of the concrete. The compressive strength and splitting tensile strength of the steam-cured concrete after the 28th day were lower than those of standard-cured concrete, and the corresponding strength of the naturally cured concrete was always the lowest.

(2) Based on the fracture toughness of the concrete $K_{\mathrm{IC}}^{S}$, steam curing significantly improved the early fracture toughness of the concrete. However, after the 28th day, the improvement in fracture toughness of the standard-cured concrete was better than that of the steam-cured concrete. The standard curing condition was more conducive to the concrete developing long-term crack resistance. Based on the critical crack-tip opening displacement ${\mathrm{CTOD}}_{c}$ from the perspective of long-term performance, when the concrete crack was about to develop, the critical crack opening value of the standard-cured concrete was greater than that of the steam-cured concrete. The allowable deformation value of the steam-cured concrete was smaller, demonstrating that steam curing can significantly increase the brittleness of concrete. Based on the critical strain energy release rate $G_{\mathrm{IC}}^{S}$, the ability of standard-cured concrete to resist crack propagation was better than the steam-cured concrete, and steam curing weakens the ability of concrete to resist crack propagation.

(3) The material length Q and characteristic length $l_{\mathrm{ch}}$ of the concrete decreased with age. Before the 7th day, the Q and $l_{\mathrm{ch}}$ of the steam-cured concrete were the largest. After the 28th day, the Q and $l_{\mathrm{ch}}$ of the standard-cured concrete were the largest. The Q and $l_{\mathrm{ch}}$ of the naturally cured concrete were at a minimum at each age. Compared to the standard curing condition, the Q and $l_{\mathrm{ch}}$ of the steam-cured concrete decreased faster. With respect to the concrete developing long-term brittleness, the steam curing condition will increase the brittleness of the concrete. From the overall trend, the tension–compression strength ratio decreased with age, which was the same as the trend observed for Q and $l_{\mathrm{ch}}$. However, from the data before the 28th day, the development pattern of the tension–compression strength ratio could not be obtained. The brittleness parameters such as material length Q and characteristic length $l_{\mathrm{ch}}$ were more suitable for evaluating the brittleness of the concrete.

(4) A time-dependent model of concrete fracture toughness that considered the curing condition and loading age was established. Based on the comparisons between the model predicted value and the test values that were obtained from other studies in the literature, the error is acceptable. The proposed model presents strong adaptability and is able to accurately predict the development of concrete fracture toughness.

## Figures and Tables

**Figure 1 materials-14-07865-f001:**
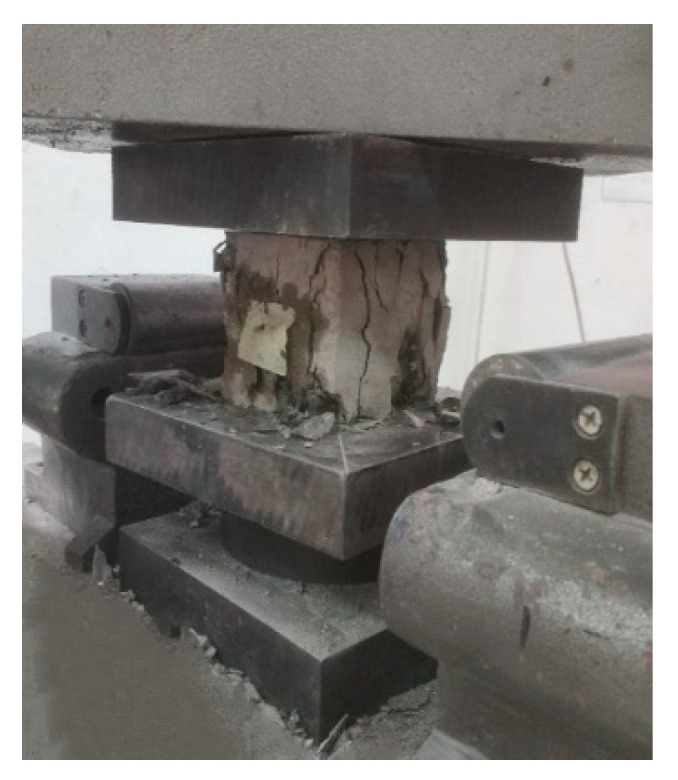
Compression strength test.

**Figure 2 materials-14-07865-f002:**
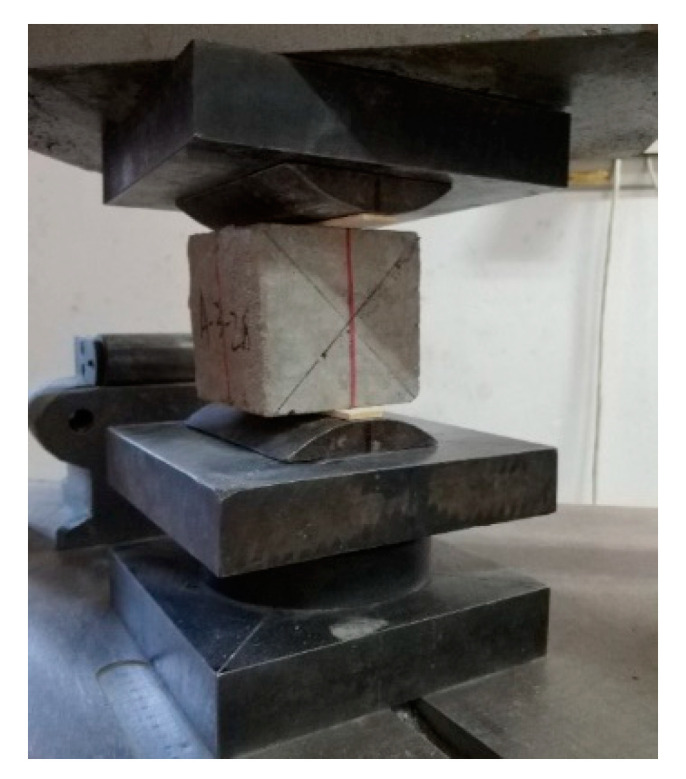
Splitting tensile strength test.

**Figure 3 materials-14-07865-f003:**
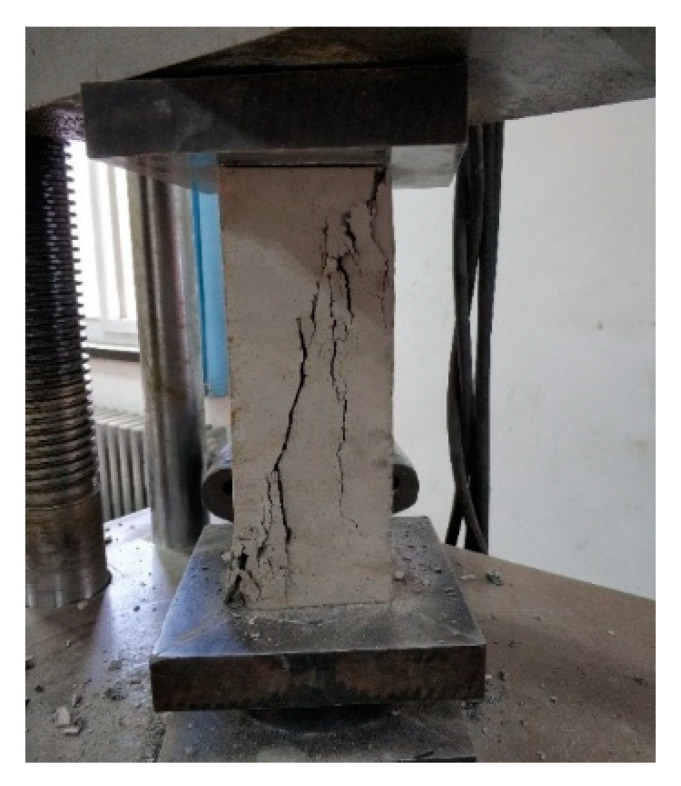
Concrete axial compression test.

**Figure 4 materials-14-07865-f004:**
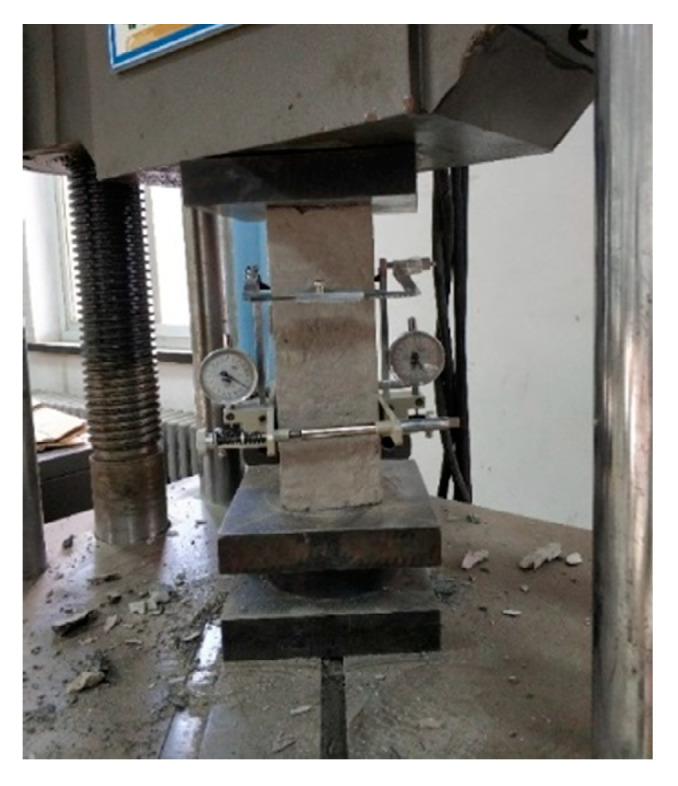
Concrete elastic modulus test.

**Figure 5 materials-14-07865-f005:**
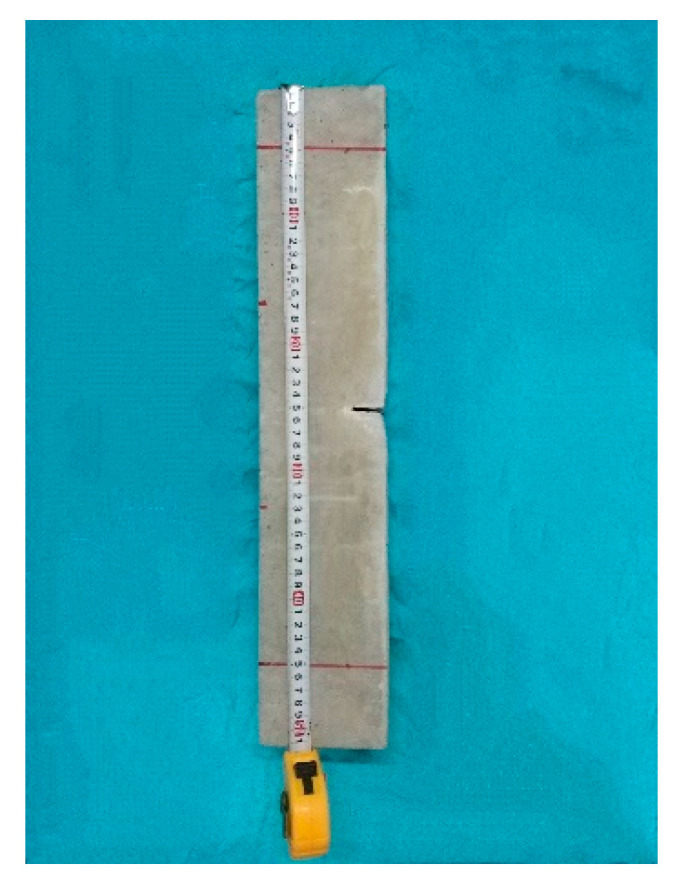
Precast notched beam.

**Figure 6 materials-14-07865-f006:**
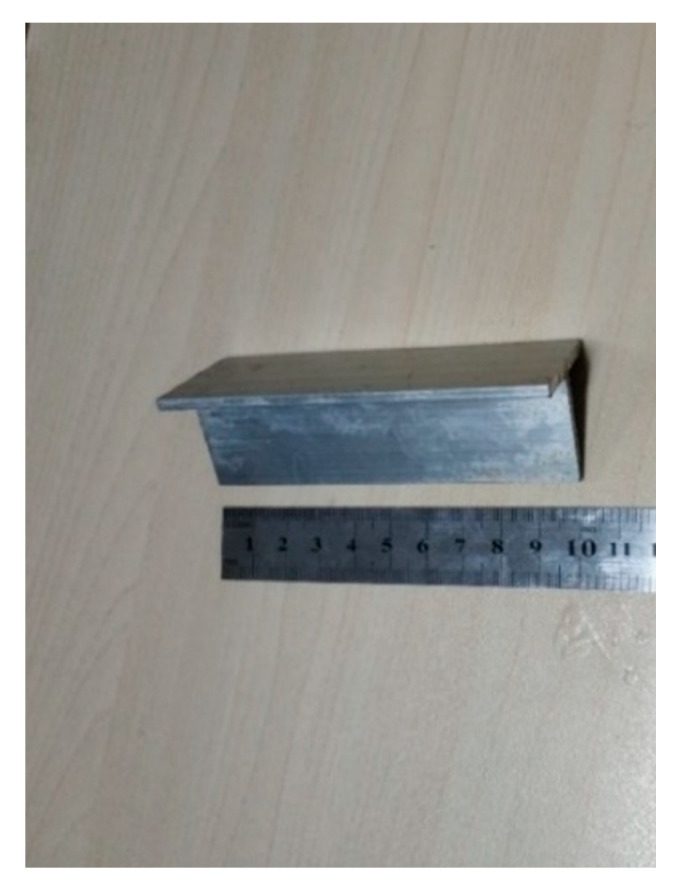
T-shaped steel plate.

**Figure 7 materials-14-07865-f007:**
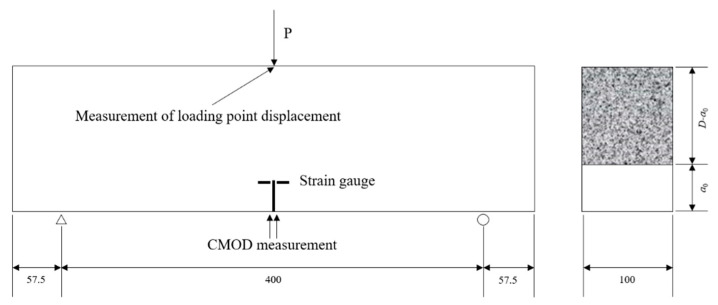
Experimental set-up and dimensions for the three-point bending test.

**Figure 8 materials-14-07865-f008:**
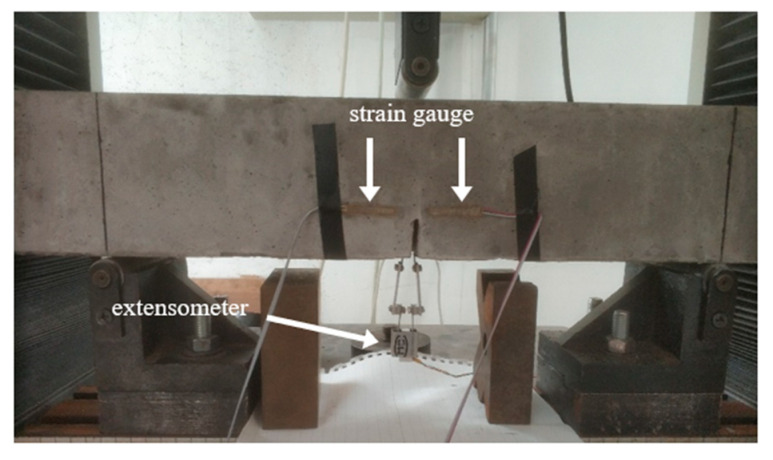
Three-point bending test setup.

**Figure 9 materials-14-07865-f009:**
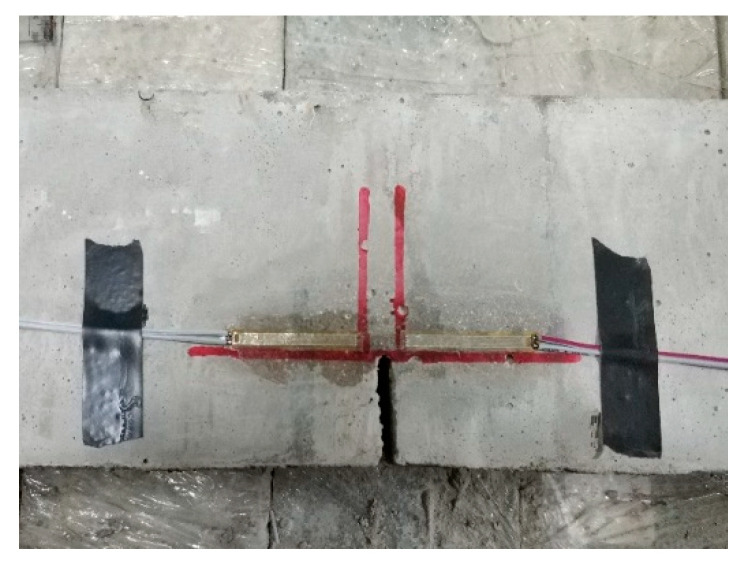
Strain gauge arrangement.

**Figure 10 materials-14-07865-f010:**
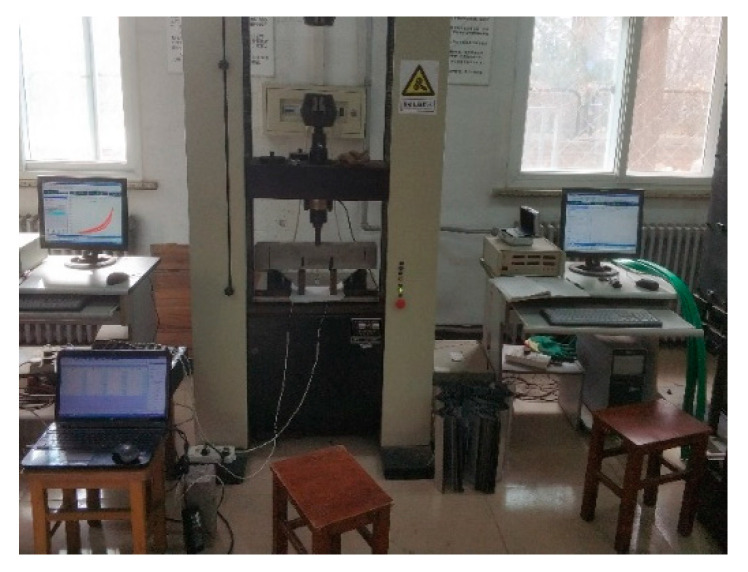
Fracture property test.

**Figure 11 materials-14-07865-f011:**
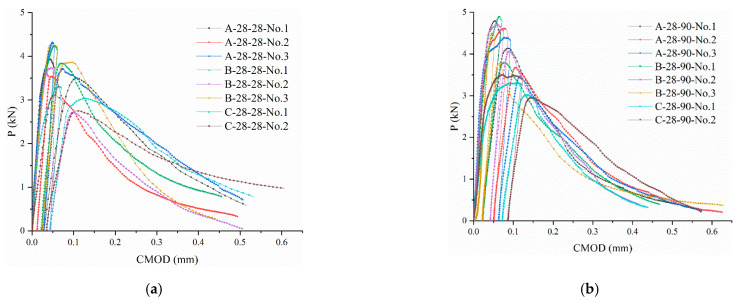
P-CMOD curves. (**a**) P-CMOD curves for A-28-28, B-28-28, and C-28-28. (**b**) P-CMOD curves for A-28-90, B-28-90, and C-28-90.

**Figure 12 materials-14-07865-f012:**
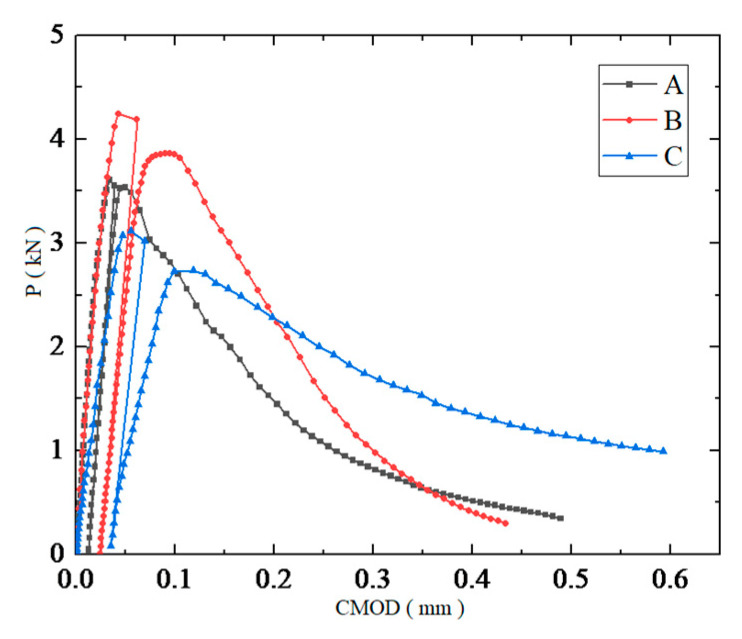
P-CMOD curves of the cured concrete unloaded.

**Figure 13 materials-14-07865-f013:**
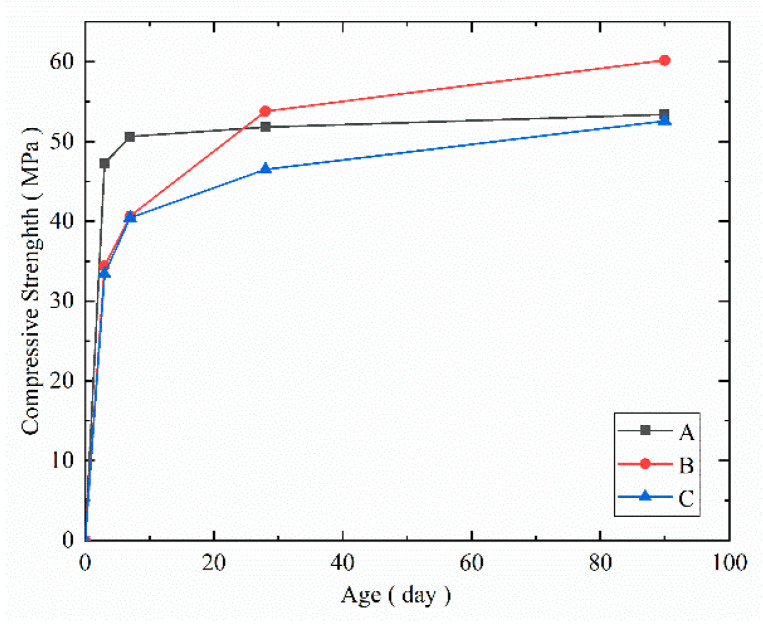
Compressive strength of the cured concrete at each age.

**Figure 14 materials-14-07865-f014:**
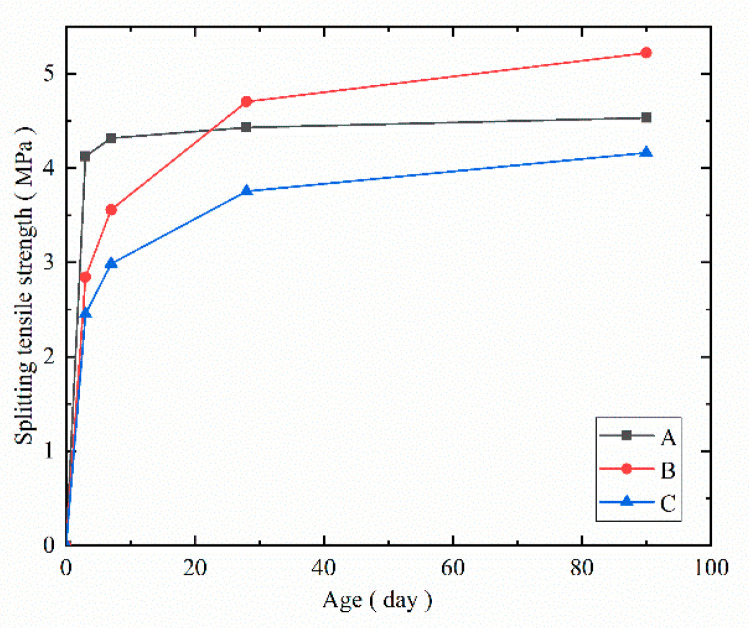
Splitting tensile strength of the cured concrete at each age.

**Figure 15 materials-14-07865-f015:**
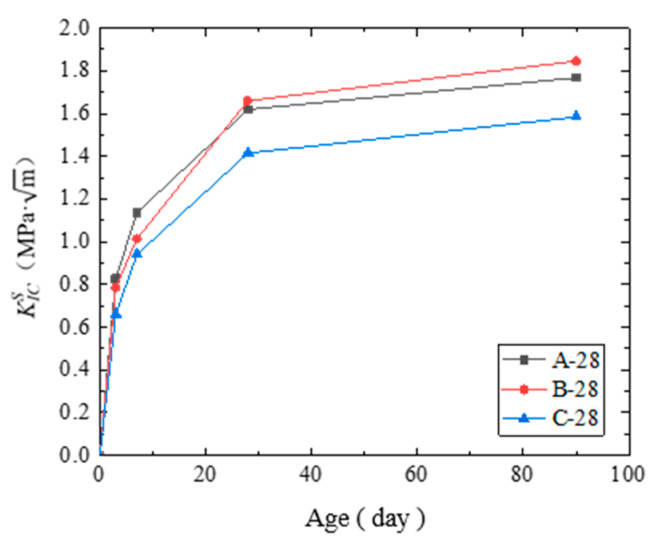
Fracture toughness $K_{\mathrm{IC}}^{S}$ of the cured concrete at each age.

**Figure 16 materials-14-07865-f016:**
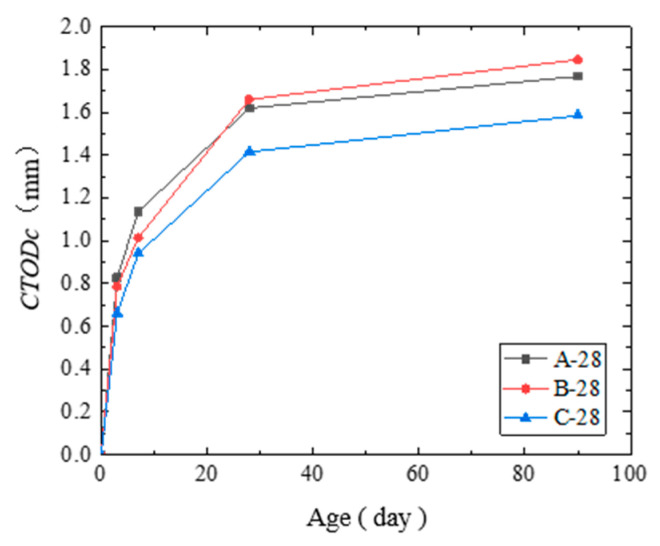
Critical crack tip opening displacements${\;\mathrm{CTOD}}_{c}$ of the cured concrete at each age.

**Figure 17 materials-14-07865-f017:**
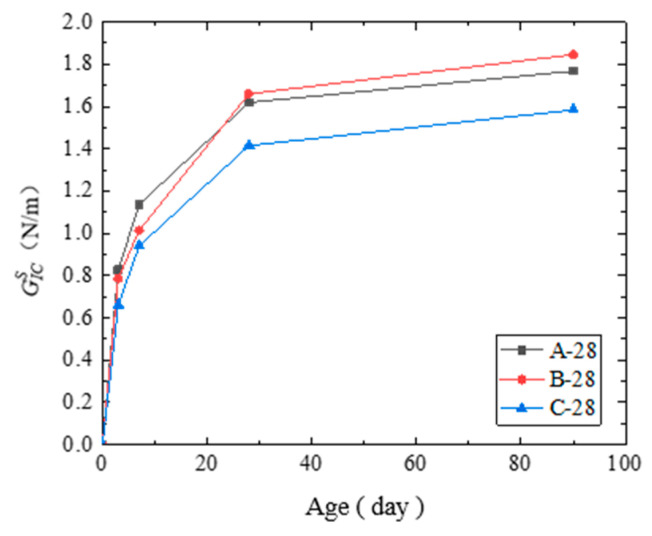
Critical strain energy release rate $G_{\mathrm{IC}}^{S}$ of the cured concrete at each age.

**Figure 18 materials-14-07865-f018:**
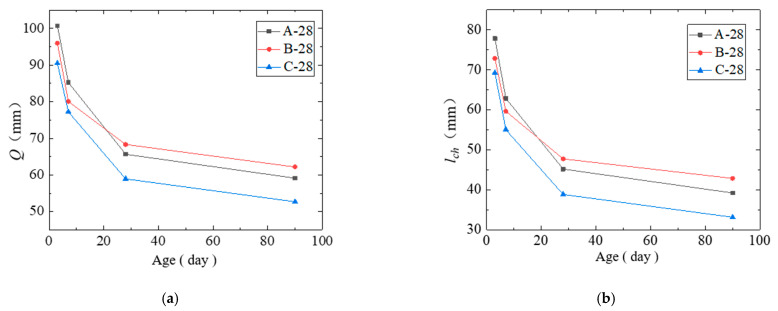
Brittleness parameters of the cured concrete at each age. (**a**) Material length Q. (**b**) Characteristic length $l_{\mathrm{ch}}$.

**Figure 19 materials-14-07865-f019:**
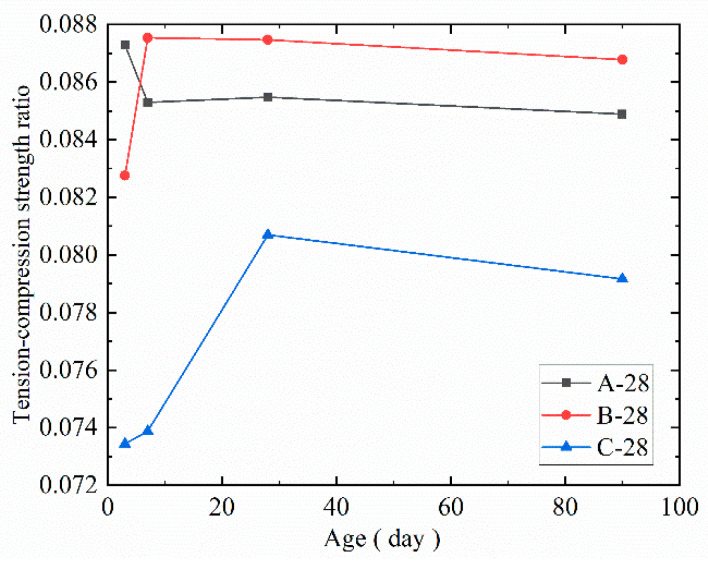
Tension–compression strength ratios of the cured concrete at each age.

**Figure 20 materials-14-07865-f020:**
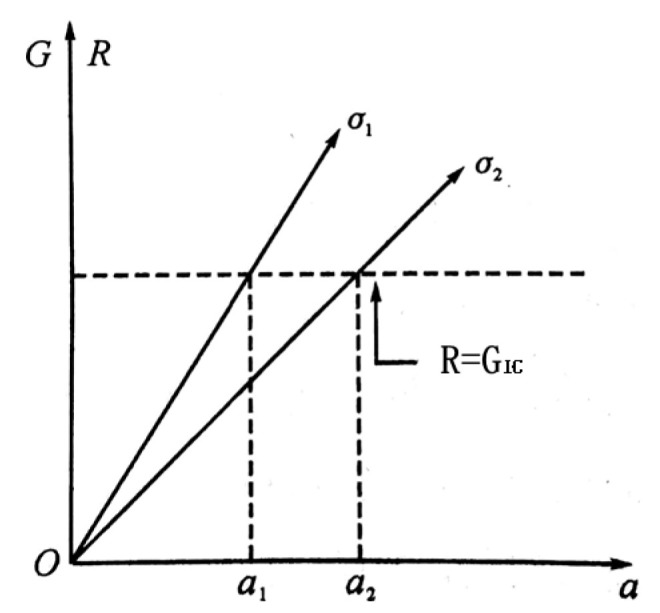
Relationship between R, G, and a.

**Figure 21 materials-14-07865-f021:**
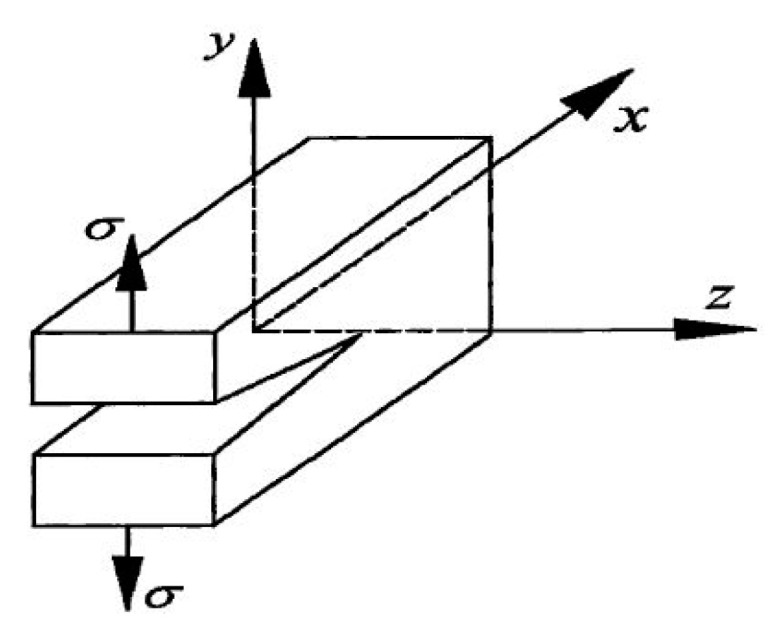
Crack calculation model of a mode I crack.

**Figure 22 materials-14-07865-f022:**
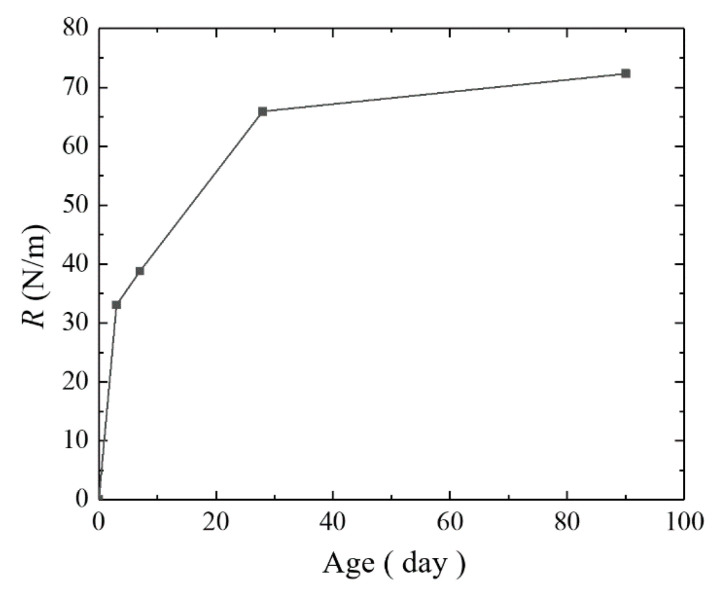
Curve of R of the steam-cured concrete loaded at the 3rd day.

**Figure 23 materials-14-07865-f023:**
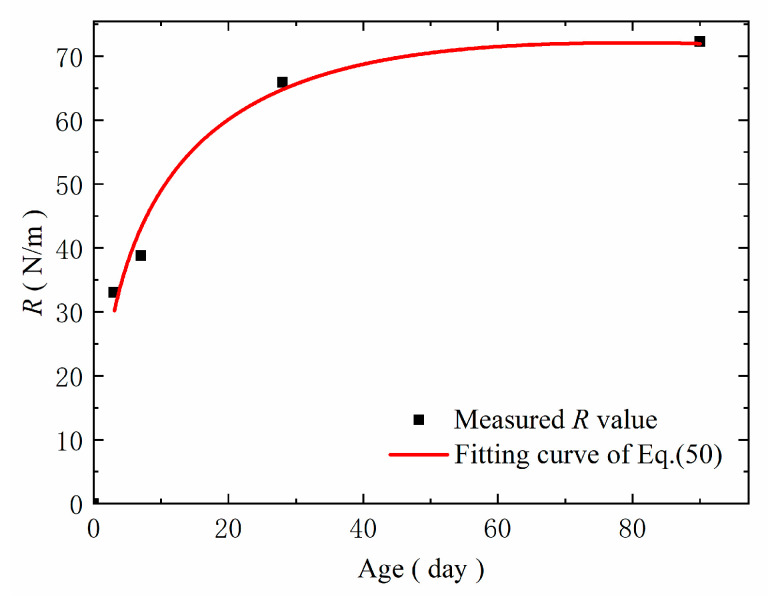
Fitting curve of concrete crack propagation resistance R.

**Figure 24 materials-14-07865-f024:**
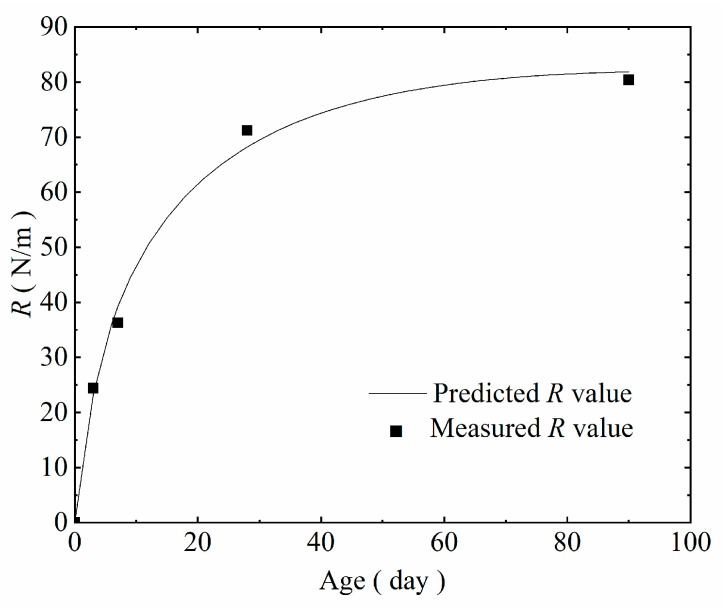
Comparison of model curves and the measured values.

**Table 1 materials-14-07865-t001:** Physical properties of the ordinary Portland cement (OPC).

Materials	Standard Consistency Water Consumption (%)	Stability	Setting Time (h)	
			Initial Setting	Final Setting
OPC	27.7	qualified	2.2	3.7

**Table 2 materials-14-07865-t002:** Chemical composition of the ordinary Portland cement (OPC).

Materials	CaO	SiO_2_	Al_2_O_3_	Fe_2_O_3_	MgO	Na_2_O	K_2_O	MnO	TiO_2_	P_2_O_5_	SO_3_
OPC	63.66	22.42	6.11	4.34	0.92	0.22	0.55	0.14	0.23	0.05	0.26

**Table 3 materials-14-07865-t003:** Chemical composition of the fly ash.

Materials	CaO	SiO_2_	Al_2_O_3_	Fe_2_O_3_	MgO	Na_2_O	K_2_O	Cl	TiO_2_	P_2_O_5_	SO_3_	NiO
Fly ash	4.01	53.96	31.15	4.16	1.01	0.89	2.04	0.13	1.13	0.67	0.73	0.11

**Table 4 materials-14-07865-t004:** Concrete mix proportions.

W/B (%)	W (kg/m^3^)	OPC (kg/m^3^)	Fly Ash (kg/m^3^)	FA (kg/m^3^)	CA (kg/m^3^)	SP (kg/m^3^)
0.3	147	392	98	688	1075	1.96

**Table 5 materials-14-07865-t005:** Specimen Groups.

Group	Age of Loading (Days)	Age of Testing (Days)	Testing Method and Number of Specimens		
			Splitting Tensile Strength	Compressive Strength	Fracture Properties
A-3	3	3, 7, 28, 90	12	12	12
A-7	7	7, 28, 90	9	9	9
A-28	28	28, 90	6	6	6
B-28	28	3, 7, 28, 90	12	12	12
C-28	28	3, 7, 28, 90	12	12	12
Total			51	51	51

**Table 6 materials-14-07865-t006:** Crack initiation load and load limit.

Number	$\mathrm{Crack}\;\mathrm{Initiation}\;\mathrm{Load}\;P_{\mathrm{ini}}\;(\mathrm{kN})$	$\mathrm{Load}\;\mathrm{Limit}\;P_{\max}\;(\mathrm{kN})$	$P_{\mathrm{ini}}$ $/P_{\max}$
A-3-3	3.26	3.68	0.885
B-28-3	2.23	2.61	0.855
C-28-3	1.88	2.08	0.901
A-3-7	3.65	4.06	0.898
A-7-7	3.70	4.16	0.890
B-28-7	3.31	3.83	0.863
C-28-7	2.50	2.73	0.916
A-3-28	3.80	4.15	0.916
A-7-28	3.82	4.23	0.903
A-28-28	3.55	3.95	0.898
B-28-28	3.58	4.08	0.877
C-28-28	2.95	3.21	0.920
A-3-90	4.17	4.53	0.922
A-7-90	4.29	4.61	0.930
A-28-90	4.30	4.60	0.936
B-28-90	4.26	4.70	0.907
C-28-90	3.20	3.40	0.942

**Table 7 materials-14-07865-t007:** Compressive and splitting tensile strengths of the cured concrete at each age.

Test	Curing Condition	Age of Test (Day)			
		3	7	28	90
Compressive strength (MPa)	Steam curing	47.26	50.62	51.82	53.40
	Standard curing	34.37	40.65	53.79	60.17
	Natural curing	33.42	40.42	46.52	52.58
Splitting tensile strength (MPa)	Steam curing	4.12	4.32	4.43	4.53
	Standard curing	2.84	3.56	4.70	5.22
	Natural curing	2.45	2.99	3.75	4.16

**Table 8 materials-14-07865-t008:** Fracture toughness $K_{\mathrm{IC}}^{S}$ of the cured concrete at each age.

Curing Condition	Age of Test (day)	$K_{\mathrm{IC}}^{S}$ $\;(\mathrm{MPa}\cdot \sqrt{m})\;$			Mean	S.D.	C.V.
Steam curing	3	0.776	0.891	0.813	0.827	0.048	5.78%
	7	--	1.147	1.127	1.137	0.010	0.90%
	28	1.524	1.617	1.718	1.620	0.079	4.88%
	90	1.707	1.892	1.705	1.768	0.088	4.97%
Standard curing	3	0.910	0.758	0.693	0.787	0.091	11.59%
	7	1.030	0.954	1.059	1.014	0.044	4.35%
	28	1.664	1.847	1.470	1.660	0.154	9.28%
	90	1.821	2.085	1.632	1.846	0.186	10.07%
Natural curing	3	0.750	0.604	0.629	0.661	0.064	9.64%
	7	0.919	1.059	0.842	0.940	0.090	9.58%
	28	1.404	1.425	--	1.415	0.011	0.77%
	90	1.671	1.498	--	1.585	0.087	5.46%

**Table 9 materials-14-07865-t009:** Critical crack tip opening displacements${\;\mathrm{CTOD}}_{c}$ of the cured concrete at each age.

Curing Condition	Age of Test (Day)	${\mathrm{CTOD}}_{c}\;(\mathrm{mm})\;$			Mean	S.D.	C.V.
Steam curing	3	0.0114	0.0109	0.0123	0.0115	0.0005	4.75%
	7	--	0.0108	0.0141	0.0124	0.0017	13.41%
	28	0.0132	0.0131	0.0130	0.0131	0.0001	0.58%
	90	0.0150	0.0145	0.0153	0.0150	0.0003	2.14%
Standard curing	3	0.0107	0.0099	0.0098	0.0101	0.0004	4.09%
	7	0.0115	0.0111	0.0126	0.0117	0.0006	5.38%
	28	0.0131	0.0136	0.0135	0.0134	0.0002	1.52%
	90	0.0169	0.0154	0.0151	0.0158	0.0008	4.90%
Natural curing	3	0.0110	0.0085	0.0103	0.0099	0.0010	10.48%
	7	0.0108	0.0116	0.0097	0.0107	0.0008	7.39%
	28	0.0120	0.0120	--	0.0120	0.0000	0.15%
	90	0.0121	0.0138	--	0.0130	0.0009	6.83%

**Table 10 materials-14-07865-t010:** Critical strain energy release rate $G_{\mathrm{IC}}^{S}$ of the cured concrete at each age.

Curing Condition	Age of Test (day)	$G_{\mathrm{IC}}^{S}\;(N/m)$			Mean	S.D.	C.V.
Steam curing	3	30.60	33.82	34.78	33.06	1.79	5.40%
	7	--	42.11	40.68	41.40	0.71	1.73%
	28	68.73	68.05	72.50	69.76	1.96	2.81%
	90	76.08	78.59	76.99	77.22	1.04	1.34%
Standard curing	3	23.08	23.67	26.46	24.40	1.47	6.03%
	7	38.28	34.96	35.63	36.29	1.43	3.94%
	28	71.35	74.27	68.10	71.24	2.52	3.54%
	90	80.77	81.58	78.87	80.41	1.13	1.41%
Natural curing	3	24.85	22.93	20.83	22.87	1.64	7.18%
	7	34.21	35.52	32.34	34.02	1.30	3.83%
	28	65.41	61.40	--	63.40	2.00	3.16%
	90	69.51	71.12	--	70.32	0.81	1.14%

**Table 11 materials-14-07865-t011:** Brittleness parameters of the cured concrete at each age.

Curing Condition	Brittleness Parameter (mm)	Age of Test (Day)			
		3	7	28	90
Steam curing	Q	100.77	85.24	65.64	59.13
	$l_{\mathrm{ch}}$	77.84	62.85	45.18	39.21
Standard curing	Q	96.01	80.06	68.33	62.17
	$l_{\mathrm{ch}}$	72.85	59.64	47.74	42.86
Natural curing	Q	90.52	77.22	58.93	52.67
	$l_{\mathrm{ch}}$	69.22	55.04	38.84	33.16

**Table 12 materials-14-07865-t012:** Tension–compression strength ratios of the cured concrete at each age.

Curing Condition	Age of Test (Day)			
	3	7	28	90
Steam curing	0.0873	0.0853	0.0855	0.0849
Standard curing	0.0828	0.0875	0.0875	0.0868
Natural curing	0.0734	0.0739	0.0807	0.0792

**Table 13 materials-14-07865-t013:** Development of R of the steam-cured concrete loaded at the 3rd day.

Age (Day)	0	3	7	28	90
$G_{\mathrm{IC}}^{S}$ (N/m)	0	33.06	38.80	65.90	72.31

**Table 14 materials-14-07865-t014:** The known parameter values in Equation (50).

Parameter	s	RH	$\alpha_{E}\;$	$f_{cu,k}$	$\sigma \left(t_{0}\right)$	h	$t_{0}$
Value	0.2	40%	0.9	50 MPa	1.959 MPa	50 mm	3

**Table 15 materials-14-07865-t015:** The values of the unknown parameter in Equation (50).

Parameter	α	β	$\mathrm{Adj}.{\;R}^{2}$
Value	0.24386	−0.00354	0.99096

**Table 16 materials-14-07865-t016:** Comparison of calculated R values and the measured R results of the standard-cured concrete.

Age (Day)	3	7	28	90
Model Predicted R value (N/m)	23.61	38.34	67.38	81.57
Measured R value (N/m)	24.40	36.29	71.24	80.41
Error rate	−3.24%	5.65%	−5.42%	1.46%

**Table 17 materials-14-07865-t017:** Comparison of model predicted values and reference of [30].

Age (Days)	1	2	7	28
Measured $K_{\mathrm{IC}}^{S}$ (MPa·$\sqrt{m}$)	0.85	1.12	1.30	1.54
Model Predicted $K_{\mathrm{IC}}^{S}$ (MPa·$\sqrt{m}$)	0.77	0.94	1.24	1.58
Error rate	−10%	−16%	−5%	3%

**Table 18 materials-14-07865-t018:** Comparison of model predicted values and reference of [35].

Curing Temperature (°C)	5	20	40	60
Measured $K_{\mathrm{IC}}^{S}$ (MPa·$\sqrt{m}$)	1.33	1.46	1.65	1.61
Model Predicted $K_{\mathrm{IC}}^{S}$ (MPa·$\sqrt{m}$)	1.21	1.55	1.72	1.81
Error rate	−9.40%	6.30%	4.11%	12.57%

## Data Availability

Not applicable.

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
