# Peer review of "Brittleness of Concrete under Different Curing Conditions"

_materials, 2021, doi:10.3390/ma14247865_

Round 1
Reviewer 1 Report
An experimental study of the steam curing on the static strength of concrete in comparison with standard curing and natural curing is carried out. The authors of the article propose a model that predicts the observed patterns.
1) In the penultimate paragraph in the introduction, an overly general overview is made concerning general questions of fracture mechanics. Of the links provided in this paragraph, the two links 2003 and 2009 are the most recent. The rest are 1976-1991. This fact raises doubts in the conclusion that there are no researchers involved in the formulation of a model of hardening from the number of days. It is also not said that there is a standard for determining the static strength of concrete that sets the optimal number of days for concrete hardening.
2) Confused by the term time-dependent model. The term time-dependent is too general a term and can refer not only to static load, but also to dynamic or usefulness. It would be nice to use the name of the model curing time-dependent model. In fig. 11 a c samples 1 and 3 show similar values for modes A-28-28 and B-28-28. Thus, steam curing is not inferior to standard curing, although the authors noticed a difference only in 2 samples.
3) In standard processing, the strength is often measured at 28 days. The question arises for the authors, in which standards it is necessary to measure the strength before 28 days. If there are no such standards for measuring strength, then where can the benefits of the small number of days aging shown in Figures 13-19 be exploited? Please add a couple of sentences in the introduction explaining the practical importance of steam curing time of concrete if the standard method shows higher strength with longer hardening times.
Reviewer 2 Report
The topic of the paper is interesting and important from practical point of view.
The experimental program and obtained test results are presented lengthily enough.
However, the remarks below are provided in order to improve the paper.
- English language should be elevated. There are some grammar errors and mistakes, particularly in the chapter “Introduction” (for example terminology: it would be better to use “researchers” instead of “scholars”).
- The proposed model was calibrated due to the authors test results and a good agreement between calculated and experimental results was obtained. The fit of the model was much lower when verifying it on the basis of test data taken from other experiments. This fact should be discussed and the conclusion should be added on the possibility of putting the model into practice.
Reviewer 3 Report
This study performed an experimental program to determine the effects of curing conditions and loading ages on the concrete brittleness. Different mechanical tests were considered for this investigation. The paper is well-written and well-organized. The reviewer recommends some comments before publishing, as follows:
- Abstract: please add quantitative results at the end of the abstract.
- Page 1, line 16: steam method of curing is not based on a standard? Please explain about standard curing? Is it like the water curing method?
- Page 1, lines 37-39: Please use reference for this sentence.
- Page 2, line 48: please use names of authors instead of “scholars”.
- Page 2, line 61: use researchers instead of scholars.
- Page 2, line 61: please do not use scholars and use the names.
- Section 2.2: please mention the density of cement and aggregates. Also, please mention the slump values of mixtures.
- Page 6, lines 186-187: please do not use a short paragraph throughout the manuscript.
- Fig. 11: please compare results in 1 or 2 curves instead of showing them separately.
- Page 7, lines 199-200: please do not use a short paragraph.
- Page 7, line 202: please explain the standard curing.
- Conclusions: please provide a paragraph before concluding remarks to explain the experimental program briefly.
Round 2
Reviewer 3 Report
Dear Editor,
The authors appropriately improved the manuscript structure.
Best Regards,
Seyed Sina Mousavi